# Transformation of dolutegravir into an ultra-long-acting parenteral prodrug formulation

Suyash Deodhar [1,9], Brady Sillman [1,2,9], Aditya N. Bade[1], Sean N. Avedissian [3], Anthony T. Podany[3], JoEllyn M. McMillan[1], Nagsen Gautam[4], Brandon Hanson[1], Bhagya L. Dyavar Shetty[1], Adam Szlachetka[2], Morgan Johnston[1], Michellie Thurman[1], Daniel J. Munt[5], Alekha K. Dash[5], Milica Markovic [1], Arik Dahan [6], Yazen Alnouti[4], Alborz Yazdi[7], Bhavesh D. Kevadiya[1], Siddappa N. Byrareddy [1], Samuel M. Cohen [8], Benson Edagwa [1,2,7✉] & Howard E. Gendelman [1,2,4,7✉]

Ultra-long-acting integrase strand transfer inhibitors were created by screening a library of monomeric and dimeric dolutegravir (DTG) prodrug nanoformulations. This led to an 18-carbon chain modified ester prodrug nanocrystal (coined NM2DTG) with the potential to sustain yearly dosing. Here, we show that the physiochemical and pharmacokinetic (PK) formulation properties facilitate slow drug release from tissue macrophage depot stores at the muscle injection site and adjacent lymphoid tissues following single parenteral injection. Significant plasma drug levels are recorded up to a year following injection. Tissue sites for prodrug hydrolysis are dependent on nanocrystal dissolution and prodrug release, drug-depot volume, perfusion, and cell-tissue pH. Each affect an extended NM2DTG apparent half-life recorded by PK parameters. The NM2DTG product can impact therapeutic adherence, tolerability, and access of a widely used integrase inhibitor in both resource limited and rich settings to reduce HIV-1 transmission and achieve optimal treatment outcomes.

[1] Department of Pharmacology and Experimental Neuroscience, University of Nebraska Medical Center, Omaha, NE 68198, USA. [2] Nebraska Nanomedicine Production Plant, University of Nebraska Medical Center, Omaha, NE 68198, USA. [3] Department of Pharmacy Practice and Science, University of Nebraska Medical Center, Omaha, NE 68198, USA. [4] Department of Pharmaceutical Sciences, University of Nebraska Medical Center, Omaha, NE 68198, USA. [5] Department of Pharmacy Sciences, Creighton University, Omaha, NE 68178, USA. [6] Department of Clinical Pharmacology, Ben-Gurion University of the Negev, Beer-Sheva 84105, Israel. [7] Exavir Therapeutics, Inc., New York, NY 10012, USA. [8] Department of Pathology and Microbiology, University of Nebraska Medical Center, Omaha, NE 68198, USA. [9] These authors contributed equally: Suyash Deodhar, Brady Sillman. ✉email: benson.edagwa@unmc.edu; hegendel@unmc.edu

Vaccination remains the principal means to prevent viral infections enabling the elimination of smallpox, measles, polio, and rubella[1–4]. However, such success has not been achieved for the human immunodeficiency virus type one (HIV-1). Despite four decades of research, complete prevention of HIV-1 transmission has not been achieved by vaccination. Viral suppression and pre-exposure prophylaxis (PrEP) were realized only by antiretroviral therapy (ART)[5]. While ART treatment of HIV-1-infected or susceptible persons reduced morbidity and mortality, signs and symptoms of infection continue to coordinate with low-level viral replication[6,7]. ART remains the gold standard of treatment for people living with HIV-1 (PLWH). Nonetheless, regimen limitations in drug compliance, toxicities, and tolerability affect viral drug resistance[8,9]. Treatment cessation leads to viral rebound coincident with co-morbid cancers and opportunistic infections[10–12]. This includes viral hepatitis where chronic antiviral therapy is mandated[13]. Thus, while ART profoundly improves life quality and longevity for PLWH, therapeutic limitations remain[14].

The major deterrent to ART efficacy is the lack of regimen adherence, linked to the social stigma of storing and taking daily medicines, along with depression and substance abuse disorders. These also affect HIV-1 transmission rates[15–17]. Such concerns have ushered in an era of long-acting (LA) parenteral ART. LA ART drugs currently approved or under investigation include cabotegravir (CAB), rilpivirine (RPV), and lenacapavir among others[18–22]. Each is designed for therapeutic and PrEP applications[22–24]. However, injection site reactions, administration volume, drug–drug interactions, resistance, and the required monthly to bimonthly parenteral drug administration limit currently approved LA ART use present therapeutic challenges[25–27]. As such, efforts have shifted towards the development of ultra-long-acting (XLA) ART that can maintain efficacious plasma drug concentrations through extended dosing intervals, such as quarterly, every six months, or once yearly.

In this work, we have now achieved such a drug dosing interval by creating a DTG prodrug encased in a surfactant-coated nanocrystal (coined as NM2DTG). NM2DTG significantly extends the apparent half-life of parent DTG and produces a unique flattened plasma PK profile for an XLA dosing regimen. Microscopy and spectroscopy studies affirm the stable and unique prodrug-nanoparticle composition. DTG half-life extensions are mediated by prodrug release rates from the nanocrystal linked to its hydrophobicity, unique physiochemical properties, and prodrug cleavage rates. These are linked to the pH, injection volume, tissue perfusion, and protein and lipid composition. These biochemical and pharmacological events underly DTG's transformation to an XLA. The impact of the reported NM2DTG formulation rests in its utility to prevent HIV-1 transmission and adherence to drug regimens where extended dosing can significantly affect disease prevention and treatment outcomes[28].

## Results

**Prodrug synthesis and physicochemical characterizations.** We now demonstrate that optimal ester carbon lengths can yield substantive changes to the pharmacological properties of LA antiretroviral drugs (ARVs)[29,30]. The attachment of DTG on one side of an 18-carbon fatty acid chain promoiety through an ester linkage markedly extends the drug's apparent half-life when compared to other prodrug monomers of varied chain lengths or an 18 carbon DTG dimer (Fig. 1A). DTG and its prodrugs were characterized by Fourier-transformation infrared (FT-IR) spectroscopy, nuclear magnetic resonance (NMR), and electrospray ionization mass spectrometry (ESI-MS) (Fig. 1B and Supplementary Figs. 1–5). Altogether, the results show that the attachment of

variable carbon chains to the parent drug affects both aqueous and octanol solubility (Fig. 1C, D) and is dependent on the hydrocarbon chain length. However, M4DTG, with an 18-carbon fatty acid with DTG molecules attached on both ends, exhibited higher aqueous solubility than the 18-carbon lipid bearing a single DTG attachment. This also resulted in the octanol solubility reflective of the parent drug. To determine whether the chemical modifications influenced antiviral activity, the half-maximal inhibitory concentration ($IC_{50}$) of the prodrugs was tested in human MDM challenged with HIV-1$_{ADA}$. HIV-1 reverse transcriptase (RT) activities from each of the treatment groups demonstrated comparable $IC_{50}$ values for DTG, MDTG, and M2DTG (Fig. 1F; 2.4, 3.2, and 3.1 nM, respectively). Such modifications also elicited stable drug-to-polymer interactions through increased hydrophobicity and lipophilicity, limiting degradation of the nanoparticles and drug dissolution from the solid nanocrystal matrix. Nanoformulations of DTG (NDTG), MDTG (NMDTG), and M2DTG (NM2DTG) were generated by high-pressure homogenization. XRD confirmed the crystalline form of the drug nanoformulations, with NMDTG and NM2DTG having similar diffraction patterns divergent from that of NDTG (Supplementary Fig. 6). The thermal properties and physical states of the prodrugs and their respective nanoformulations were studied by differential scanning calorimetry (DSC) (Fig. 1E and Supplementary Fig. 7) and thermogravimetric analysis (TGA) (Supplementary Fig. 8). The DSC thermograms affirmed content uniformity and lack of thermally distinct polymorphs. The TGA thermograms showed that the drugs (Supplementary Fig. 8A) and nanoformulations (Supplementary Fig. 8B) had no residual solvents and were stable across temperatures of 30-300 °C. The physical stability of NDTG and NM2DTG was monitored by measuring particle hydrodynamic diameter (size), polydispersity (PDI), and zeta potential during storage by dynamic light scattering (DLS; Fig. 1G, H and Supplementary Fig. 9). Particle size and PDI was unchanged over 265 days. The stability of the nanoformulations was investigated by parent drug and or prodrug quantification over 301 days during room temperature storage (Supplementary Fig. 9C). The nanoformulations showed consistent drug levels with limited evaporation of the NM2DTG formulation.

**Macrophage uptake, retention, and antiretroviral activities.** NM2DTG was readily taken up by MDM at both 5 and 25 μM treatment concentrations during a 24 h evaluation (Supplementary Fig. 11A and Fig. 2A; respectively). Dose-dependent differences in uptake were observed, with 13.9 and 75.5 nmol M2DTG/$10^6$ cells following 5 or 25 μM treatment at 24 h. Higher drug uptake was seen for NM2DTG when compared against NMDTG. For each drug concentration, free drug and nanoformulation showed no cytotoxicity by MTT tests (Supplementary Fig. 10). Following 8 h drug loading, MDM retained significantly higher amounts of NM2DTG at both 5 and 25 μM concentrations over a 30-day test period (Supplementary Fig. 11B and Fig. 2B; respectively). NMDTG was near baseline at day 10 for both doses, while M2DTG levels were consistent over 30 days. Dose-dependent differences in retention were also observed, with 5.7 and 21.07 nmol M2DTG/$10^6$ cells present at day 30 following 5 or 25 μM treatment. NDTG yielded little to no uptake or retention of drug in MDM at any recorded time point. TEM visualized intracellular particles after treatment with 25 μM NM2DTG for 8 h in MDM (Supplementary Fig. 12). NM2DTG was observed in intracellular MDM vesicles immediately after treatment and extending to day 30. These data sets affirmed the recorded retention drug levels. The antiretroviral activities of the DTG prodrug nanocrystals were evaluated after a single 8 h administration of 1 or 10 μM to MDM followed by a challenge with HIV-1$_{ADA}$ at 5-day intervals

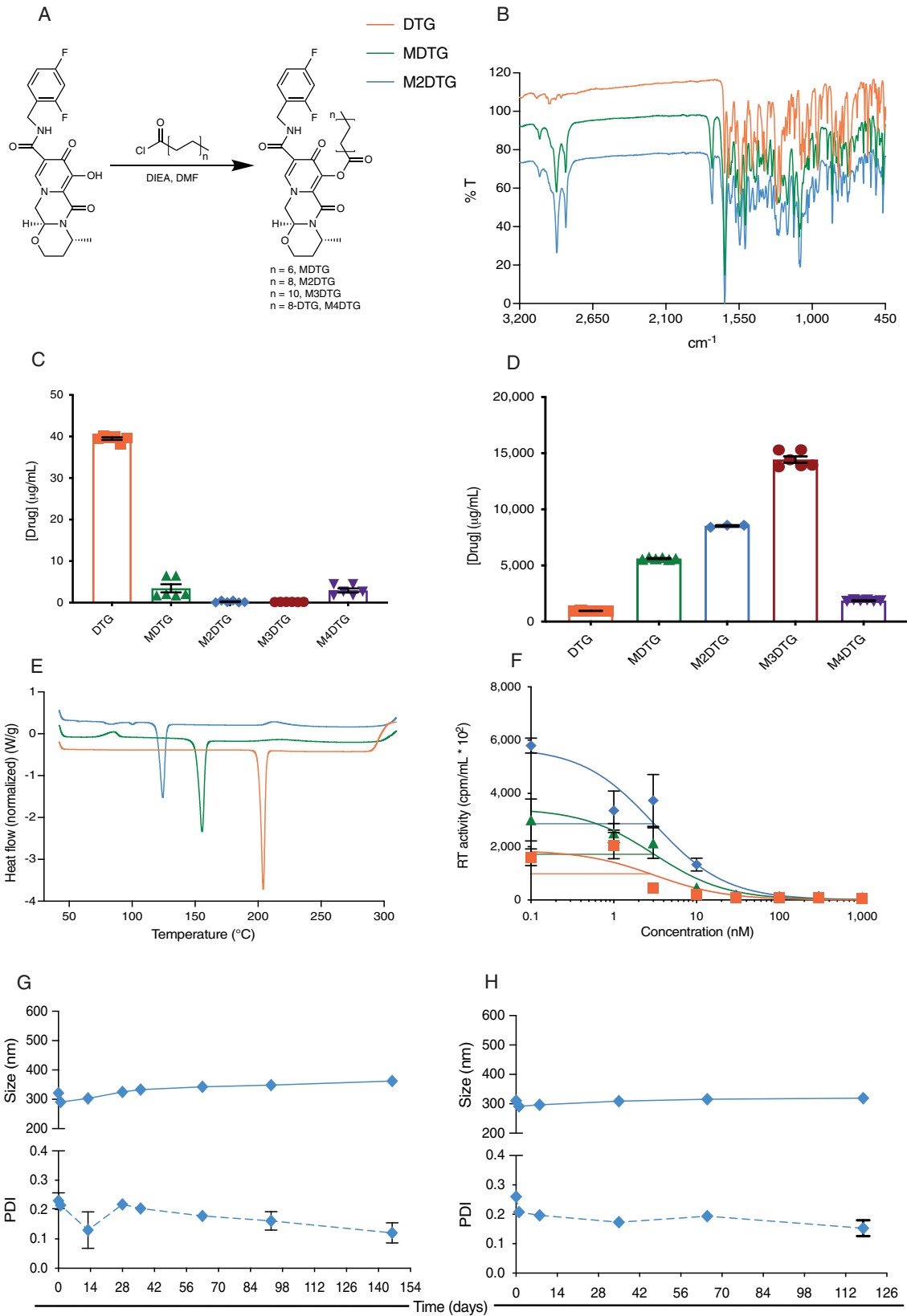

to day 30. Viral infection was assessed by HIV-1p24 antigen expression in cells by immunocytochemical staining (Fig. 2C). NDTG showed evident viral replication beginning at day 5 at both 1 and 10 μM treatments, while NM2DTG protected MDM against HIV-1 challenge for 30 days at both drug concentrations.

**Pharmacokinetics (PK).** PK tests were performed in male Balb/cJ mice, male SD rats, and female RMs. Male Balb/cJ mice were administered a single 45 mg DTG-eq./kg (equimolar DTG) dose of NDTG, NMDTG, NM2DTG, NM3DTG, or NM4DTG intramuscularly (IM) into the caudal thigh muscle to determine PK

**Fig. 1 Synthesis and characterization of DTG prodrugs. A** DTG was synthesized by esterification of the DTG hydroxyl group yielding lipophilic prodrugs with 14, 18, or 22-carbon chains. They are named MDTG, M2DTG, and M3DTG. A DTG dimer with an 18-carbon chain ester linkage was named M4DTG. **B** FT-IR overlays of DTG (orange), MDTG (green), and M2DTG (blue) identified the presence of specific molecular groups in the prodrugs affirmed by spectroscopy. **C** Aqueous and **D** octanol solubility of DTG and prodrugs are illustrated. Results are expressed as the mean ± SEM for $N = 3$ independent replicates quantified in duplicate. **E** Differential scanning calorimetry (DSC) thermogram overlays of DTG, MDTG, and M2DTG determined changes in physical state. **F** The antiretroviral half-maximal inhibitory concentration ($IC_{50}$) in macrophages was assessed at 0.1–1000 nM concentrations by measurements of HIV-1 reverse transcriptase (RT) activity in culture supernatants. Results are expressed as the mean ± SEM for $N = 4$ biological replicates. **G, H** Particle stability of high concentration NM2DTG nanoformulations for potential clinical translation, including the hydrodynamic diameter (size; solid lines) and polydispersity index (PDI; dashed lines), were tested at 22 °C up to 146 days by dynamic light scattering (DLS). **G** NM2DTG formulated with 38% (w/v) M2DTG, Polysorbate 20 and $PEG_{3350}$ achieved a final concentration of 268 mg M2DTG/mL. **H** NM2DTG formulated with 45% (w/v) M2DTG, Polysorbate 20 and $PEG_{3350}$ achieved a final concentration of 330 mg M2DTG/mL. Results are expressed as the mean ± SEM for $N = 3$ independent replicates. Source data are provided in the Figshare database under Digital Object Identifier (DOI) code https://doi.org/10.6084/m9.figshare.19026452.

profiles during a one-year observation period. Plasma samples were analyzed by UPLC-MS/MS to assess parent drug levels (Fig. 3A). NM2DTG displayed a significantly reduced DTG decay curve compared to any of the other formulations, with plasma drug levels dropping below the protein-adjusted $IC_{90}$ (PA-$IC_{90} = 64$ ng/mL) at >367 days. Plasma levels were at or below the limit of quantitation (LOQ = ~1 ng/mL) at day 35 for NDTG, day 287 for NMDTG, and day 210 for NM4DTG, while they remained above the PA-$IC_{90}$ for the entire test period for NM2DTG (69.6 ng/mL at day 367). DTG apparent half-life increased from 3.48 days for NDTG and 39.1 days for NMDTG, to 167.9 days for NM2DTG in plasma of mice after receiving single-drug formulation injections (Supplementary Table 2).

Male SD rats were administered a single 45 mg DTG-eq./kg dose of NDTG or NM2DTG intramuscularly (IM) into the caudal thigh muscle to determine PK over 1 year. Plasma samples were analyzed by UPLC-MS/MS to assess parent drug levels (Fig. 3B). NM2DTG displayed a significantly reduced DTG decay curve compared to NDTG in mice and rats. Plasma levels were at or near the LOQ = ~1 ng/mL at day 42 for NDTG (1.1 ng/mL at day 42), while they remained above the PA-$IC_{90}$ until day 308 for NM2DTG (77.4 ng/mL at day 308). Peak plasma M2DTG levels were at 4 hours (100.8 ng/mL), but only detectable at day 7 (1.4 ng/mL) where they fell below the LOQ (Fig. 3B insert). Tissue biodistribution was assessed at days 57, 175, and 364 for both prodrug (M2DTG) and parent drug (DTG) in the muscle injection site, spleen, lymph nodes (pooled; cervical, axillary, and inguinal), and liver (Fig. 3D–G). Parallel drug measurements were done in the lung, gut, kidney, rectum, and brain (Supplementary Fig. 14A–E). Tissue drug levels following NDTG treatment were limited, with a maximum of 66.7 ng/g in lymph nodes (pooled) at day 57. Tissue drug levels following NM2DTG were notably higher, with measurable parent and prodrug at all time points in all tissues apart from the brain. Liver and spleen exhibited higher levels of prodrug than the parent drug, while lymph nodes (pooled) had similar levels at each time point. Kidney showed prodrug at or slightly above baseline measurement levels. Significant active drug levels were recorded in the kidney and likely reflect native drug excretion. Lymph nodes (pooled), lungs and rectum showed the highest drug levels. The highest prodrug levels were detected at the site of injection and sustained throughout the year-long study. Taken together, the data sets indicate that the muscle represented the primary drug depot where the nanocrystals are either slowly absorbed or dissociated releasing the prodrug which is then converted into the active native drug. Neither NDTG nor NM2DTG treatments had any adverse effect on animal weights or metabolic profiles, with no lasting differences between treatments and controls (Supplementary Fig. 13 and Supplementary Table 1). Modest metabolic differences were observed briefly after NM2DTG administration

at day 3 but rapidly resolved without lasting effects. Differences included elevated alkaline phosphatase and phosphorous, and decreased amylase and potassium. No erythema or swelling was observed at the injection site. DTG apparent half-life increased from 4.53 days for NDTG to 108.76 days for NM2DTG in rat plasma after receiving single-drug formulation injections (Supplementary Table 3).

Female rhesus macaques (RM) were administered a 45 mg DTG-eq./kg dose of NM2DTG intramuscularly (IM) into the quadriceps muscle to determine PK. An initial dose was given on day zero followed by a second booster dose on day 217. Plasma samples were analyzed by UPLC-MS/MS to determine drug levels (Fig. 4A). As observed in rodents, NM2DTG displayed a greatly reduced DTG decay curve. Following the first dose, plasma levels were at or above the PA-$IC_{90}$ until day 154 (65.3 ng/mL). Before the booster dose on day 217 plasma DTG levels fell to 45.3 ng/mL. After boosting, plasma DTG levels increased to match those seen after the first dose (974.7 vs 1102 ng/mL, respectively), however, plasma prodrug levels were 1.6 times higher following the boost than the first dose (Fig. 4A). NM2DTG exhibited an extended DTG half-life when compared against a previous study in RMs. In that study, DTG levels were 86 and 28 ng/mL on days 35 and 91 after a single NMDTG IM injection administered at 25.5 mg DTG-eq./kg[31]. In contrast, NM2DTG maintained stable plasma drug levels for the study duration.

Necropsies were performed on the NM2DTG-treated RM. To assess toxicity, we performed histological evaluation at the injection site and in solid organ, lymphoid, and brain tissues (Fig. 4B–M). All H&E staining of spleen, liver, muscle injection site, lung, ileum, lymph nodes (cervical, axillary, mesenteric, inguinal, and colonic), and brain tissue (cortex and hippocampus; Fig. 4B–M, sequentially) were reviewed by a diagnostic pathologist who affirmed that the tissues were normal without evidence of pathology.

To assess the nanocrystal drug depot and biodistribution in various tissues, we measured tissue drug levels by mass spectroscopy (MS/MS). DTG and prodrug levels were highest at the muscle injection site (17,358 and 3,802,500 ng/g, respectively) and in adjacent lymphoid tissues (9,465 and 400,601 ng/g, respectively, in inguinal lymph nodes; Fig. 4N, O). The female reproductive tract and gastrointestinal tract showed lower drug levels than proximal muscle or lymphoid tissues, but still demonstrated sustained drug levels up to day 428, and 211 days after the booster dose was given (Fig. 4P, Q).

**Prodrug hydrolysis.** Given that the activation of the ester prodrugs is mediated, in the largest measure, by enzymatic processes, we assessed the mechanisms underlying the conversion by measurements of two isoforms of carboxylesterases [(CES), CES1, and CES2]. These tests were completed in rat tissues and plasma

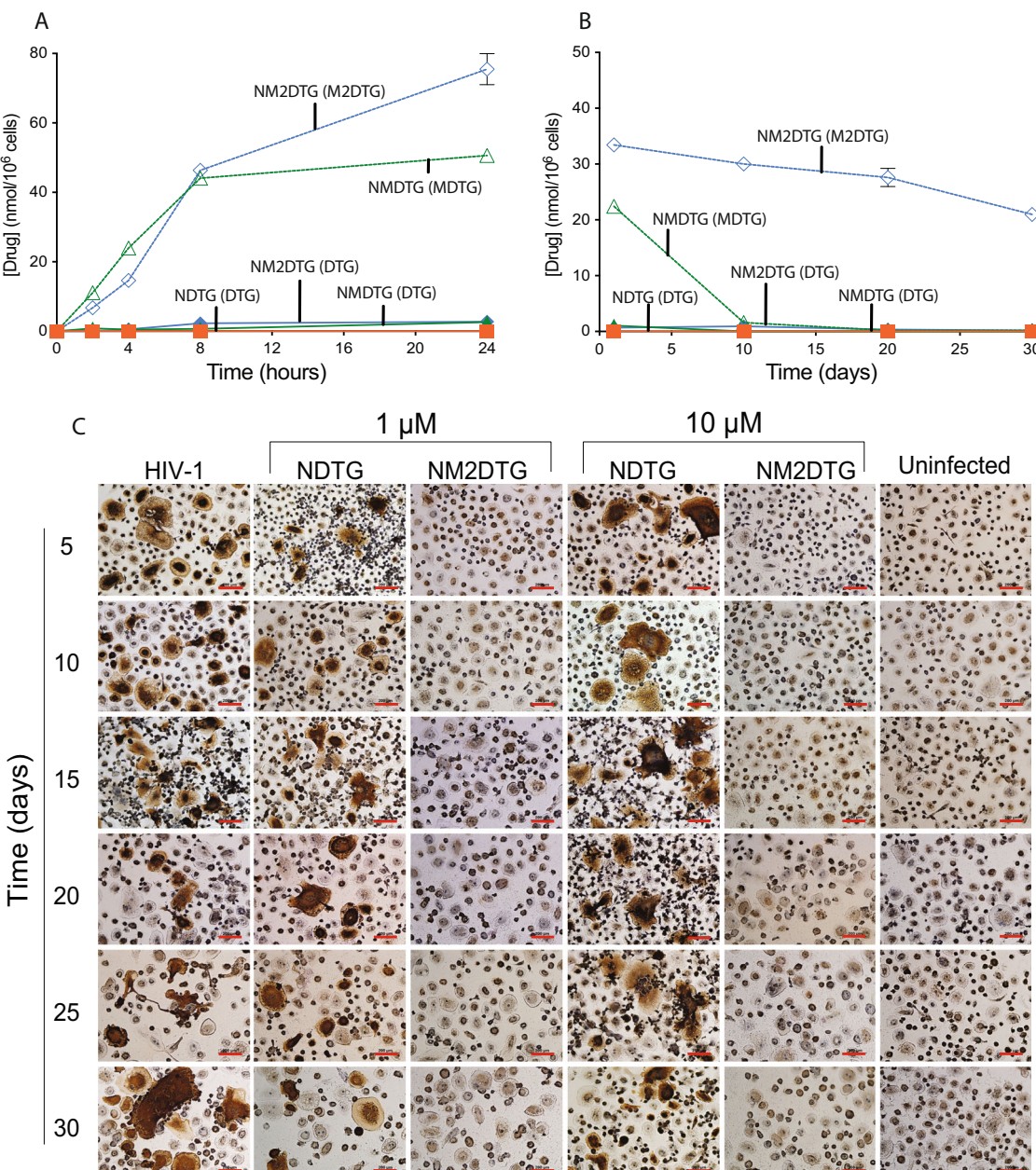

**Fig. 2 In vitro characterization of NM2DTG in MDM. A** Drug uptake and **B** retention in monocyte-derived macrophages (MDM) were measured over 24 hours and 30 days, respectively, after treatment with prodrug nanoformulations at a concentration of 25 μM. Results are expressed as the mean ± SEM for $N = 3$ biological replicates. **C** Antiretroviral responses were recorded after HIV-1$_{ADA}$ challenge at a multiplicity of infection (MOI) of 0.1 infectious virions/cell at recorded times following treatment with either NDTG or NM2DTG at 1 or 10 μM concentrations for 8 h. HIV-1p24 antigen levels were assessed in fixed MDM by immunohistochemical staining. $N = 4$ biological replicates. Representative images were taken at ×20 magnification. Scale bars —200 μm. Source data are provided in the Figshare database under Digital Object Identifier (DOI) code https://doi.org/10.6084/m9.figshare.19026983.

(Supplementary Fig. 15). Quantification of the CES isoforms demonstrated that CES1 was the most dominant. Notably, no relationships were found between CES levels and prodrug cleavage rates, indicating the process of the native prodrug to drug release was independent of a specific CES. Next, we investigated the stability of solubilized and nanoformulated M2DTG solid drug suspensions, in rat tissue homogenates, to assess time-dependent prodrug tissue cleavage. The prodrug formulation remained stable in all tissue matrices tested (Fig. 5A–H). The findings were cross-validated by quantifying parent drug levels from the same samples. The prodrug solution showed complete cleavage in the spleen and kidney homogenates at 48 h (Fig. 5C, D). For liver and muscle tissues, only

half of the prodrug solution was depleted at 48 h (Fig. 5A, E). However, the prodrug solution was rapidly cleaved in plasma, while heat-inactivation led to a significant reduction in the rates of plasma cleavage (Fig. 5G, H). Prodrug to parent drug conversion was calculated (Supplementary Fig. 17A–F). The lymph node showed the most rapid cleavage rates of the nanoformulation. Notably, the spleen and kidney showed the highest rate of prodrug cleavage when exposed to the prodrug solution. The decreasing concentration of total prodrug levels in the tissue homogenates followed second-order kinetics for all nanoformulation and most solution sample sets; with the rates obtained from the incubation of the solution in the spleen and kidney following mixed order kinetics.

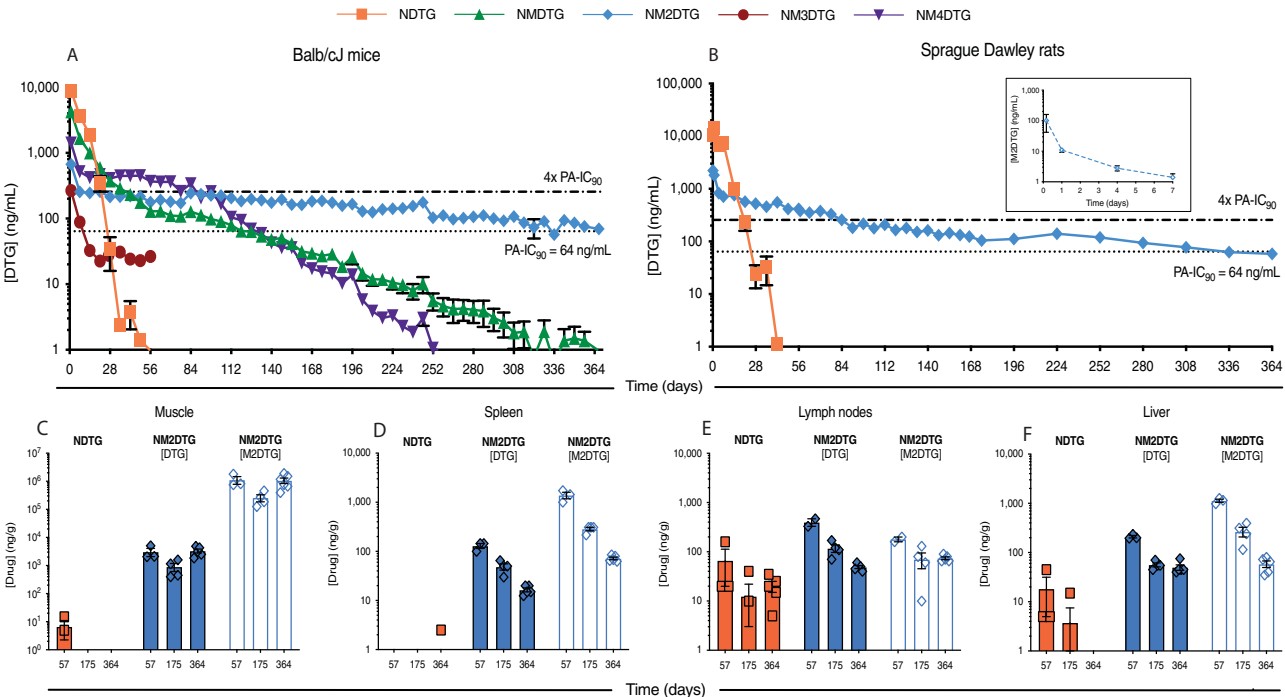

**Fig. 3 DTG PK studies in Balb/cJ mice and Sprague Dawley rats. A** Plasma DTG levels in male Balb/cJ mice administered a single intramuscular (IM) dose of NDTG, NMDTG, NM2DTG, NM3DTG, or NM4DTG. Drug formulations were injected at concentrations of 45 mg DTG-eq./kg IM in the caudal thigh and native drug levels were monitored for 367 days. Data are expressed as mean ± SEM. The study was initiated with $N = 4$–5 animals per treatment group ($N = 4$ for NM3DTG, 5 for all others). Due to loss of animals by natural causes during study period, animals at day 367 are $N = 5$ (NDTG), $N = 5$ (NMDTG), $N = 2$ (NM2DTG), $N = 4$ (NM3DTG at day 56), and $N = 5$ (NM4DTG). The dotted line indicates the DTG protein-adjusted IC$_{90}$ (PA-IC$_{90}$ = 64 ng/mL) and dashed line indicates four-times the PA-IC$_{90}$ (4× PA-IC$_{90}$ = 256 ng/mL). **B** Plasma DTG levels in male SD rats administered a single 45 mg DTG-eq./kg IM dose of NDTG or NM2DTG in the caudal thigh were recorded (solid blue diamonds and line). Plasma prodrug (M2DTG) levels are shown in the insert (open blue diamonds and dashed line). Data are expressed as mean ± SEM. The study was initiated with $N = 13$ animals per treatment group (13 per group up to day 57, 9–10 per group up to day 175, and 5–6 per group up to day 364). One animal was lost from the NDTG group on day 140 during the study period due to natural causes. **C–F** Tissue biodistribution of NDTG and NM2DTG in SD rats was assessed on days 57 ($N = 3$ animals per group), 175 ($N = 4$ animals per group), and 364 ($N = 5$ animals per group). Parent drug (DTG; solid blue diamonds) and prodrug (M2DTG; open blue diamonds) levels were determined in **C** muscle (at the site of injection), **D** spleen, **E** lymph nodes (pooled; cervical, axillary, and inguinal), and **F** liver. All drug levels were quantified by UPLC-MS/MS. Results are expressed as mean ± SEM. Exact values for each time point are provided in the Source data. Source data are provided in the Figshare database under Digital Object Identifier (DOI) code https://doi.org/10.6084/m9.figshare.19027175.

The influence of pH on the intracellular fate of the nanoformulation in macrophage endosomal microenvironments was evaluated. Tissue homogenates for the study were prepared and used for pH prodrug activation studies over 48 h (Supplementary Fig. 19). pH-dependent differences in prodrug release and hydrolysis were found linked to prodrug form, with the nanoformulation being more stable in acidic conditions than alkaline (Fig. 6A–D). The findings were cross-validated by the corresponding increase in the parent drug levels from replicate samples. The prodrug solution also displays pH-dependent hydrolysis within 48 h and is found to be more stable under acidic conditions (Fig. 6A–D). The pH-dependent prodrug activation of the nanoformulation and prodrug solution at pH 2.0, 6.0, 8.0, and 10.3 shows the profile differences linked to reaction rate kinetics (Supplementary Fig. 18A–D).

While the dissociated prodrug nanoformulation showed only 48% total prodrug left at pH 6.0 (Fig. 6B), the same formulation showed about 85% prodrug left in the spleen tissue homogenate corroborated with parent drug levels (Fig. 5C). The findings show that the nanoformulation remained stable in tissues of the same pH as the tested buffers. The reduced dissolution of the nanocrystals can explain this in the tissue matrices due to protein binding to the nanoformulation. It can thus be concluded that the

dissolution of the nanocrystals to release the prodrug is the critical factor that governs the PK outcomes.

**Histopathology and electron microscopy characterization of NM2DTG intramuscular injection.** The intramuscular delivery of NM2DTG demonstrated that the muscle was the primary drug depot site. Uninjected tissue showed normal muscle histology (Fig. 7A). Sham, saline-injected, controls showed limited macrophage responses (Fig. 7B). The histiocytic infiltration and the generation of a local injection site granulomatous reaction were induced by intramuscular delivery of NM2DTG at 45 mg DTG-eq./kg, as visualized by H&E staining in rats at day 3 after drug administration (Fig. 7C–E). Amorphous material, believed to be part of the formulation depot, was also observed in the muscle surrounded by histiocytic cells (Fig. 7C, D). The macrophages present at the NM2DTG injection site readily phagocytose the nanoformulation with intracellular storage. The low pH, in the cellular and tissue microenvironments, resulted in slow prodrug cleavage rates. H&E staining in rats at day 57 after drug administration showed a return to normal muscle histology, like that seen in the uninjected controls (Fig. 7F). Electron microscopic images of uninjected or sham controls affirm regular or

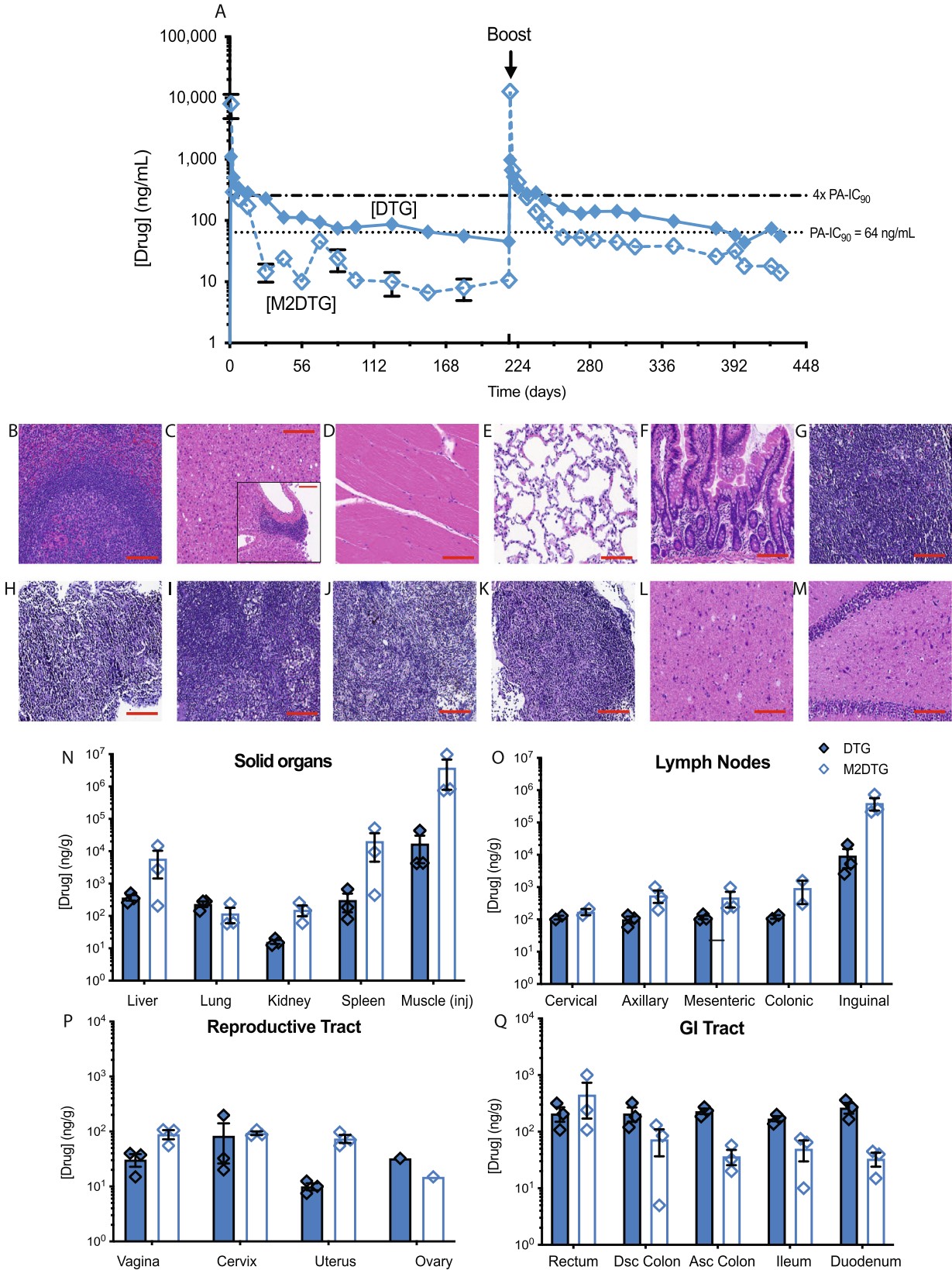

minimal changes in rat muscle histology (Fig. 7G–J). In contrast, the NM2DTG-injected rats showed extensive histiocytic infiltration with clusters of fused nanocrystals and endocytic drug particle contents (Fig. 7K, L).

This was further confirmed at the muscle site of injection in RM up to 428 days after the initial dose and following a single

booster injection on day 217 (Fig. 7M–Q). H&E staining of muscle sections at the site of injection demonstrated normal muscle histologic architecture with adjacent substantive mononuclear macrophages (Fig. 7M). There was no evidence of cellular activation or the presence of multinucleated giant cells. Transmission electron micrographs readily demonstrated clusters

**Fig. 4 DTG PK studies in rhesus macaques.** Female rhesus macaques were given a 45 mg DTG-eq./kg IM dose of NM2DTG in the quadriceps muscles, followed by an equivalent booster dose on day 217 (arrow). Animals were sacrificed on day 428 following the initial and boosted dose and tissues were collected. **A** Plasma DTG (solid blue diamonds and line) and M2DTG (open blue diamonds and dashed line) levels. Plasma samples were collected, and drug levels were determined during the experimental observation period. **B–M** H&E staining at ×20 magnification of solid tissues such as **B** spleen, **C** liver, **D** muscle, **E** lung, **F** ileum; lymph nodes (LN) such as **G** cervical LN, **H** axillary LN, **I** mesenteric LN, **J** inguinal LN, **K** colonic LN; and brain tissues such as **L** cortex, and **M** hippocampus. $N = 3$ biologically independent animals with representative images shown. Scale bars—200 μm (**B–M**). **N–Q** Tissue biodistribution of NM2DTG in rhesus macaques. Parent drug (DTG; solid blue diamonds) and prodrug (M2DTG; open blue diamonds) levels were determined in **N** solid tissues including liver, lung, kidney, spleen, and muscle (at the site of injection [inj]); **O** lymph nodes including cervical, axillary, mesenteric, colonic, and inguinal; **P** reproductive system including vagina, cervix, uterus, and ovary; and **Q** gastrointestinal (GI) tract including the rectum, descending (dsc) colon, ascending (asc) colon, ileum, and duodenum. All drug levels were quantified by UPLC-MS/MS. Results are expressed as mean ± SEM for $N = 3$ animals, except where the sample was limited. Exact values for each sample are provided in the source data. Source data are provided in the Figshare database under Digital Object Identifier (DOI) code https://doi.org/10.6084/m9.figshare.19027229.

of crystalline material, identified as NM2DTG particles, internalized into these cells (Fig. 7N–Q). Red outlines contained in the figure indicate nanocrystals present in putative endosomes (Fig. 7O).

## Discussion

Herein, an optimal novel M2DTG prodrug and its formulation were created to transform the drug's apparent half-life. The highest prodrug and drug levels were at the muscle injection site, which was sustained at a million ng/g of DTG for a year. The stability of the nanoformulations, the sustained nanocrystal dissolution, the slow tissue prodrug hydrolysis, and release from the injection site, lymphoid and solid tissues provide sustained drug seeding to blood. Rapid plasma prodrug hydrolysis concordant with sustained tissue prodrug levels parallel the extended PK profiles. The lipophilic 18-carbon promoiety on the monomeric DTG prodrug enabled its optimal physicochemical and extended PK properties. Reflective of our prior works, the transformation of antiretroviral prodrugs[29] into XLA solid drug nanoparticles requires water-insoluble compounds for compatibility with the scalable top–down nanocrystal formulation technologies. The resultant formulation needs to be crafted into optimal particle size and shape to affect dissolution, stability, cellar uptake, and drug release profile. These parameters also aid in optimizing tissue and plasma prodrug conversion kinetics to achieve sustained therapeutic drug levels at these sites. The reported improvements in the drug's PK profile also permit parallel pathways for broader prodrug transformation in drug classes targeting different parts of the viral life cycle[20]. The formulation lipophilicity facilitates enhanced drug delivery to CD4+ T cells and monocyte-macrophages, the natural HIV-1 cell reservoirs, for maximal viral suppression. LA slow effective release (LASER) ART affects drug metabolism during extended in vivo release to limit cytotoxicity[28]. The tissue pH and microenvironment, and prodrug hydrolysis were shown to be critical PK predictive parameters and may also account for the observed species differences.

Notably, other groups have also modified the structure and or delivery of DTG to develop the means to extend the drug's half-life[32–36]. In one LA biodegradable polymeric solid implants were developed by phase inversion producing removable implants[35]. However, large-scale production of optimal implants has not yet been realized. In another, palmitic acid (PA) conjugated prodrug of DTG was produced by esterification[32]. Biodegradable microparticles produced extended the drug's half-life and enabled slow drug release but formulation safety, scale-up, and long-term stability remain limitations. A third removable ultra-LA system was reported for DTG with detectable drug levels for up to 9 months but with PK variability[33].

The XLA NM2DTG formulation provides a critical link between optimal hydrophobic-lipophilic prodrug properties to extend the apparent half-life of DTG. Notably, the less hydrophobic NDTG formulation exhibits rapid native drug clearance. However, the covalent linkage of a lipophilic 18-carbon fatty acid onto DTG enhances lipophilicity and hydrophobicity to facilitate drug transport across biological barriers and slows DTG's clearance rate. Thus, we posit that aqueous solubility is one of several predictive markers for this long-acting slow effective release ART (LASER ART). Interestingly, while M2DTG and M4DTG both had 18-carbon fatty acid modification, the in vivo fates differed. Such differences underscore that the hydrophobicity and lipophilicity of the DTG prodrugs provide one, but not all, of the predictive parameters for the unique PK profiles. For example, poor dissolution of the prodrug, as demonstrated by the 22-carbon fatty ester modified NM3DTG, affected plasma DTG levels which fell below the PA-IC$_{90}$ at one week. Taken together, the current study establishes an ideal range of prodrug hydrophobicity and lipophilicity required for optimal clinical PK parameters. To this end, NM2DTG was identified as our lead candidate for further study and clinical development.

The PK boost seen during the second injection of NM2DTG in RM likely reflects an extension of the established depot present in the muscle at the site of injection, as well as within the reticuloendothelial system. The established depot in muscle represents a long-lived reservoir. Indeed, for the macrophage, the formation of granulomas at the injection site occurs consequent to cellular infiltration and uptake of the DTG prodrug nanoformulations. This notion is supported by the fact that macrophages serve both as reservoirs for persistent infection and for the drug nanoparticles[37]. In this manner, the macrophage represents a cell depot from which the prodrug is released and then hydrolyzed. In support of this notion, extensive histocyte infiltration is observed at the site of injection. Morphologically, the presence of intracellular DTG nanocrystals in macrophages characterizes the LASER ART drug depot[29,36,38–40].

We posit that the injection site within the muscle serves as a stable primary depot for the nanoformulated prodrug. Following NM2DTG parenteral injection, sustained plasma drug levels can be achieved above the PA-IC$_{90}$ in Balb/cJ mice and SD rats. Moreover, the identified prodrug profiles are based, in whole or part, on the chemical properties of the nanoformulation and its tissue penetration, perfusion, cell infiltration, and nanoparticle uptake[37,41–43]. These studies provide a starting point for human dose projection as NM2DTG is further studied and advanced towards the clinic.

The nanoformulation and the prodrug are more stable at lower pH conditions, which characterizes the macrophage subcellular microenvironments that the nanoparticles are exposed to as part of the phagocytic cascade. The retention studies of the

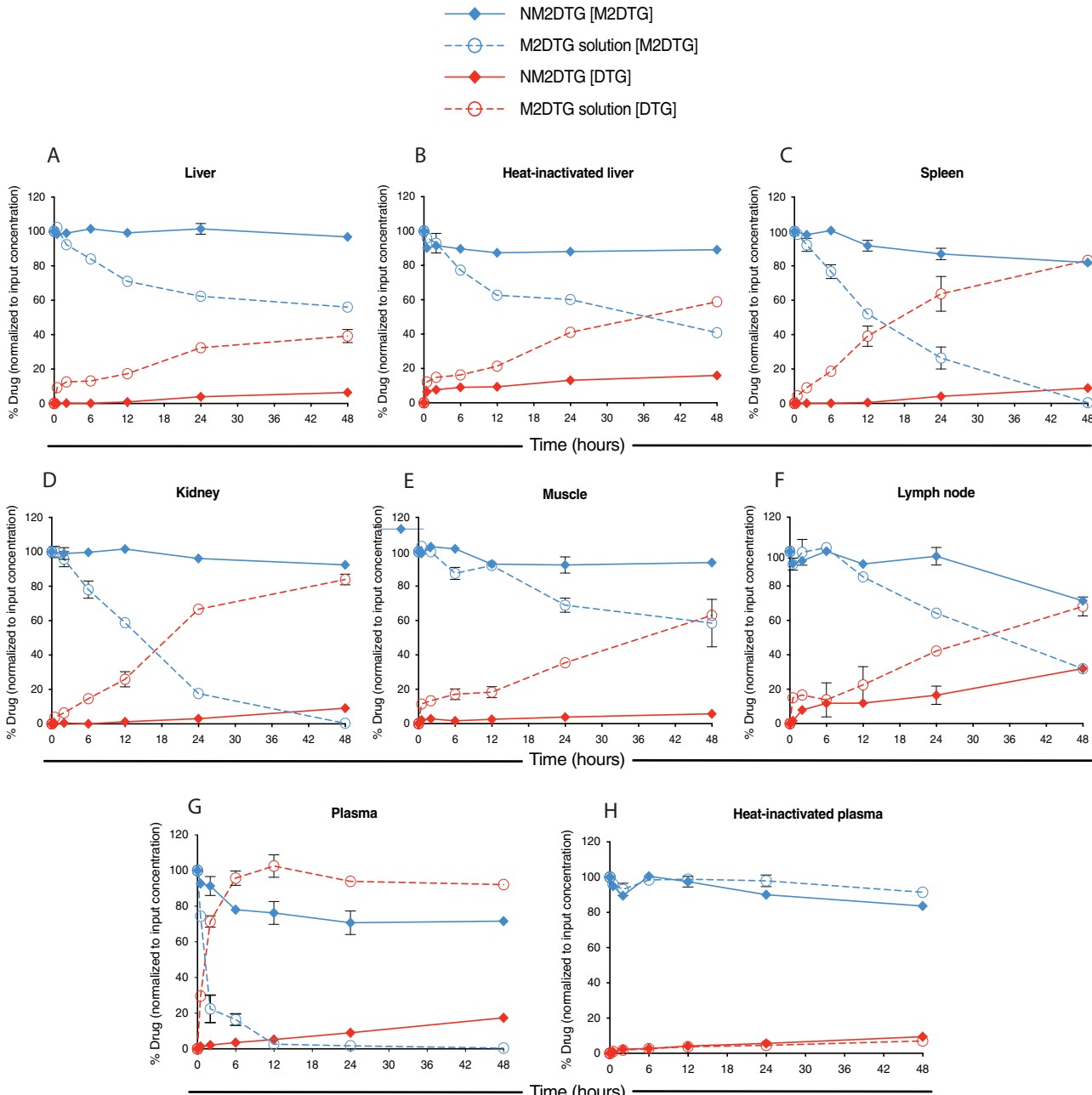

**Fig. 5 Prodrug cleavage studies in tissue drug depots.** Cleavage of M2DTG solution (dissolved in 1% (v/v) methanol; open circles) or NM2DTG nanoformulation (solid diamonds) in SD rat tissue homogenates (**A**–**F**) and plasma (**G**–**H**). Prodrug (M2DTG; blue lines) and parent drug (DTG; red lines) were quantified following M2DTG or NM2DTG incubation over 48 h in the **A** liver, **B** heat-inactivated liver, **C** spleen, **D** kidney, **E** muscle, **F** lymph node, **G** plasma, and **H** heat-inactivated plasma. All drug levels were quantified by UPLC-MS/MS. Results are expressed as mean ± SEM for $N = 3$ independent replicates ($N = 2$ for lymph nodes due to limited sample). Source data are provided in the Figshare database under Digital Object Identifier (DOI) code https://doi.org/10.6084/m9.figshare.19027253.

nanoformulation in macrophages showed persistent prodrug levels with limited concentrations of the native drug across 30 days. These observations provide insight into the slow rate of release and hydrolysis of the prodrug from the nanoformulation. Thus, intracellular accumulation of drug nanocrystals is stabilized and retained in these compartments, allowing them to release the drug into the blood slowly and as such extend the ARV's apparent half-life. The nanoformulations are stored, in measure, in lymphoid organs using these organs as secondary tissue depots. Therefore, the release of the prodrug into the extracellular matrix and plasma, resulting in its subsequent hydrolysis, underlie the

unique PK profiles of NM2DTG, as seen previously for NM2CAB[29,39].

Carboxylesterases play an important role in facilitating the activation of prodrugs. In humans, evaluation of different CES tissue isoforms shows the prominence of CES1. While the liver and kidney have the highest concentrations of CES1, those of CES2 are significantly less. However, in contrast, plasma shows no discernible levels of either of the CES species. Thus, in an attempt to better appreciate the influence of hydrolysis and the prodrug physiochemical properties, we conducted computational modeling. This was completed to simulate the enzymatic prodrug

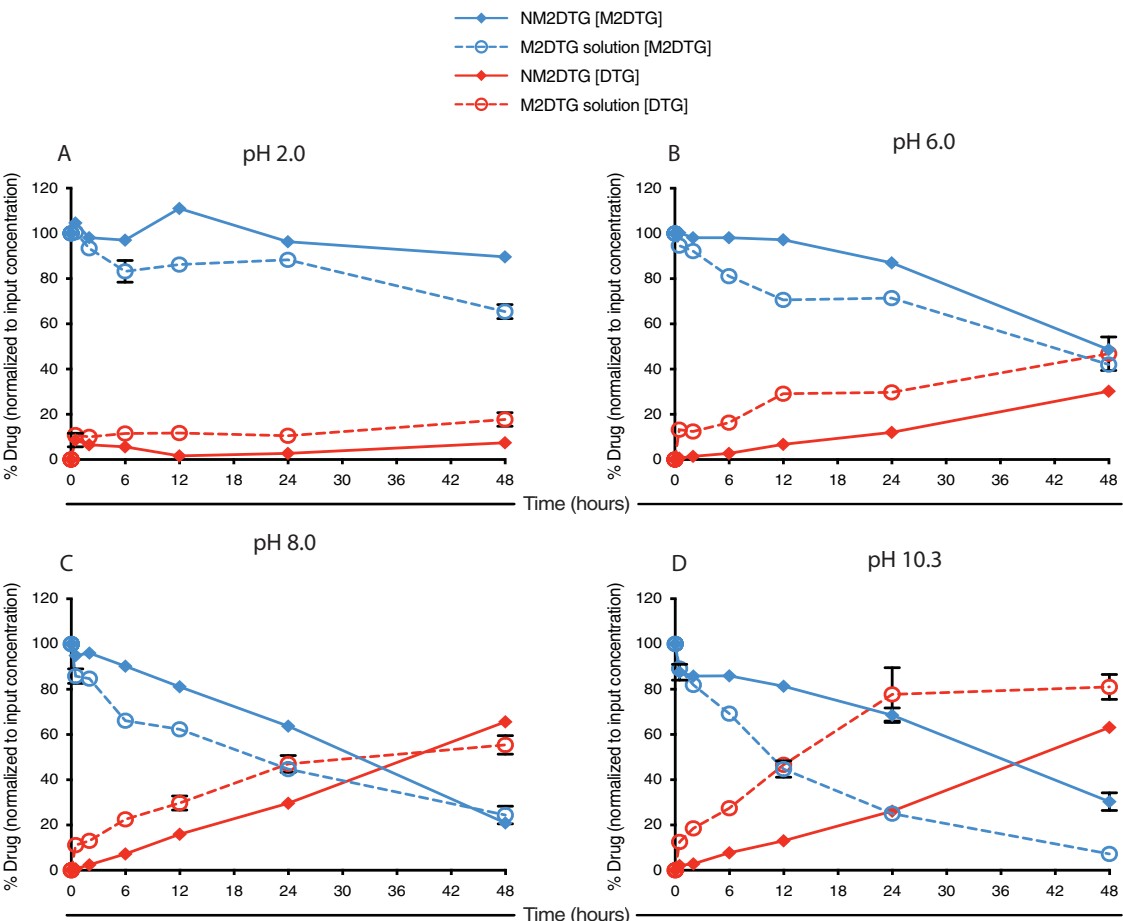

**Fig. 6 pH-dependent prodrug hydrolysis.** Hydrolysis of M2DTG solution (dissolved in 1% (v/v) methanol; open circles) or NM2DTG nanoformulation (solid diamonds) in buffers of various pH. Prodrug (M2DTG; blue lines) and parent drug (DTG; red lines) were quantified following M2DTG or NM2DTG incubation over 48 h in buffers at **A** pH 2.0, **B** pH 6.0, **C** pH 7.0, and **D** pH 10.3. All drug levels were quantified by UPLC-MS/MS. Results are expressed as mean ± SEM for $N = 3$ independent replicates. Source data are provided in the Figshare database under Digital Object Identifier https://doi.org/10.6084/m9.figshare.19362500.

hydrolysis of MDTG and M2DTG by CES. CES1 was chosen as it is known to hydrolyze ester-bond-containing drugs and is the most abundant CES enzyme[44]. MDTG, a prodrug with a shorter alkyl side chain (14-carbon units), had superior docking with the highest level of hydrolysis (Supplementary Fig. 16). The carbonyl group in MDTG faces the Ser221 residue in CES1, and the hydrophobic tail is next to a hydrophobic patch identified on the enzyme surface. Other prodrugs showed reduced hydrolysis rates that paralleled the increases in apparent half-life. Indeed, the molecular docking experiments showed that the increased length of the fatty acid chain led to decreased favorable enzyme binding of the prodrug and was linked to the PK profiles. Failure to produce immediate binding to the enzyme allows these long-side chain prodrugs to persist in biological matrices and results in slower cleavage rates. Prodrug cleavage studies the nanoformulation and free prodrug in solution do not predict PK and the actual role that each play in relationship to variant tissue environments awaits further study. This is highlighted by the failures to bridge CES levels with rates of prodrug hydrolysis. Indeed, 85% of the prodrug remained in the spleen by ex vivo tests. However, 48% of the total prodrug was seen at pH 6.0 in the solution. These studies demonstrate that the nanoformulation is stable in tissues likely due to reduced dissolution of the nanocrystals through protein-nanoformulation binding. A summative understanding points toward a cluster of variable conditions contributing toward release and hydrolysis from the prodrug formulation. A clear

delineation of prodrug stability, formulation composition and physiochemical properties is required in future studies to decipher the contribution of nanocrystal dissolution and release of free prodrug under divergent tissue and cell environments and in blood. In conclusion, the dissolution of the prodrug is a major component governing the extended PK profiles of the nanoformulated prodrug. Comprehensive dose-escalating studies will be required in future studies to determine human dosing. NM2DTG elicited plasma drug levels just above the PA-IC$_{90}$ requiring a drug boost in RMs after six months, while the murine models showed persistent plasma drug levels just above the PA-IC$_{90}$ for a year. This is in line with the previously reported terminal phase half-life of CAB in RM of 3–12 days compared to 21–50 days for humans[45,46]. However, rats have proven as a useful non-clinical model for long-acting intramuscular ester prodrug formulations, such as paliperidone palmitate[47]. Notably, the recorded sustained high prodrug and therapeutic DTG concentrations at the site of injection and lymphoid tissues following parenteral NM2DTG administration in RMs suggest that the rate of prodrug nanocrystal absorption is slower than the rate of DTG elimination. Future dose extrapolation studies that evaluate effective DTG concentrations from prodrug nanocrystals need to consider, in toto, drug depots, prodrug conversion rates, and hepatic microsomal metabolic stability.

Arguably, the most important utility of the current NM2DTG formulation rests in PrEP by providing greater access in resource-

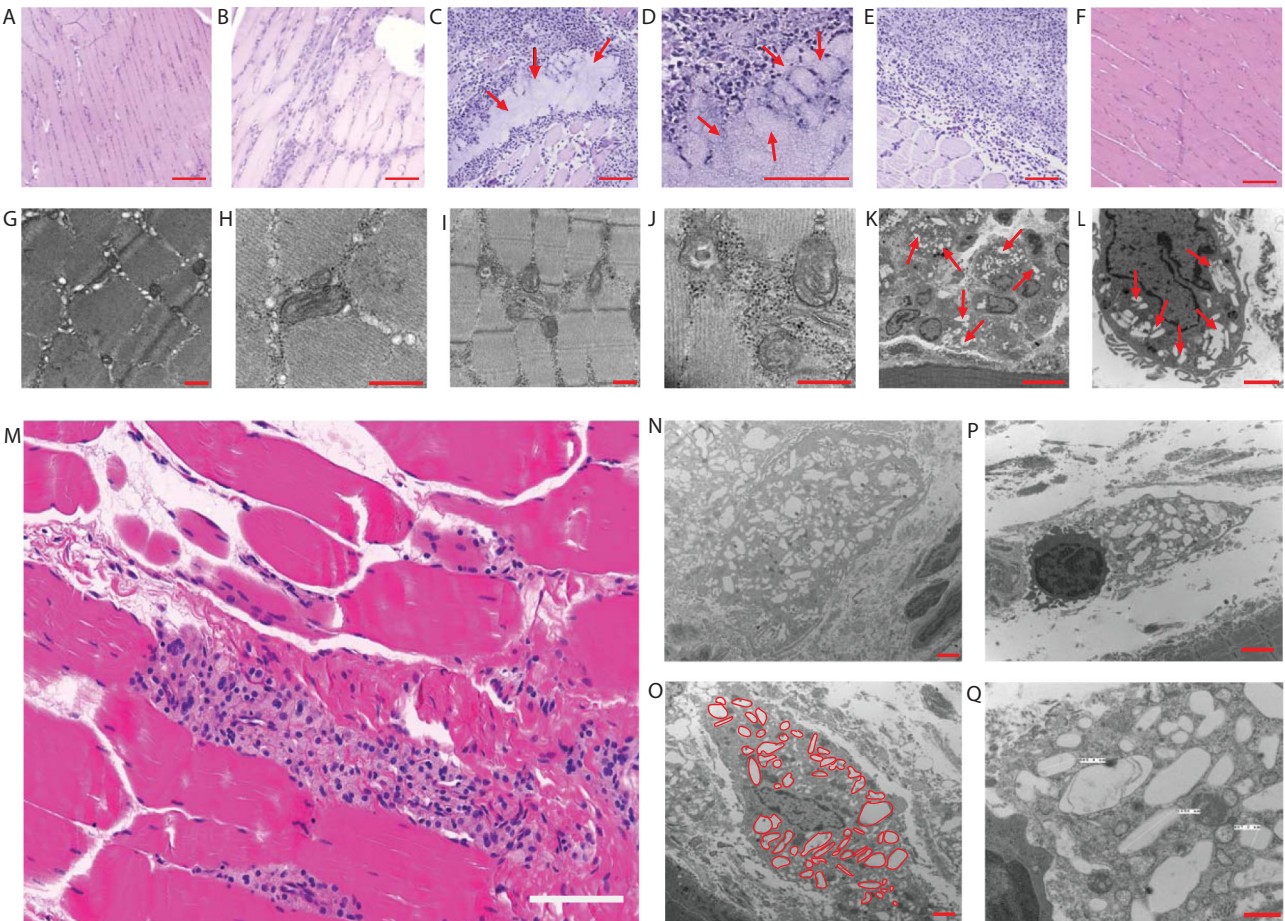

**Fig. 7 Histological and electron microscopic characterization of M2DTG nanocrystals at the injection site in SD rats and rhesus macaques. A–L** H&E staining of caudal thigh muscle from rats three days following IM injection. **A** Control (uninjected), **B** sham (saline-injected) control, and **C–E** NM2DTG (45 mg DTG-eq./kg) treated muscle sections at ×10 magnification. $N = 3$ biologically independent animals/group with representative images shown. Representative image for **C** at ×10 magnification has been provided at ×40 in **D**. **F** H&E staining of caudal thigh muscle from rats 57 days following IM injection of NM2DTG at 45 mg DTG-eq./kg at ×10 magnification. $N = 3$ biologically independent animals/group with the representative image shown. Scale bars—400 μm (**A–C**, **E**, **F**), 200 μm (**D**). **G–L** Replicate muscle samples were examined by transmission electron microscopy (TEM) from rats three days post-treatment. $N = 3$ biologically independent animals/group with representative images shown. **G**, **H** Uninjected and **I**, **J** sham (saline-injected) controls show normal muscle histology. **K**, **L** Rats that were injected with NM2DTG show cell infiltration with ingestion of the nanoformulation into endosomal vesicles (red arrows). Scale bars—500 nm (**G–J**), 10 μm (**K**), 2 μm (**L**). Female rhesus macaques were given a 45 mg DTG-eq./kg IM dose of NM2DTG in the quadriceps muscles, followed by an equivalent booster dose on day 217. Animals were sacrificed on day 428. **M** H&E staining of muscle obtained from the site of injection at ×20 magnification. **N–Q** TEM of the site of injection. Scale bars—200 μm (**M**), 2 μm (**N–P**), 500 nm (**Q**). Clusters of crystalline material identified as NM2DTG, were found to be internalized by macrophages. The red outline indicates nanocrystals present inside a macrophage that had infiltrated into the muscle. $N = 3$ biologically independent animals/group with representative images shown.

limited settings. Recent reports demonstrated that CAB LA as monotherapy is superior to daily oral tenofovir disoproxil fumarate–emtricitabine combinations in preventing HIV-1 infection in studies of men who have sex with men and transgender women[48]. NM2DTG is a particularly attractive candidate for XLA ARV therapy in both the PrEP and treatment setting, based on the safety profile and resistance patterns of parent dolutegravir, and the significant apparent half-life extension that supports extended dosing regimens. NM2DTG may also be an important component of curative strategies for HIV, in such that sustained virologic control can potentiate viral elimination[49].

To summarize, the data sets for the NM2DTG prodrug nanoformulation cartooned illustration is provided demonstrating the synthesis, injection, biodistribution, hydrolysis, and extended PK parameters (Fig. 8). The illustration was provided to track the unique aspects of the formulation design and application. Notably, such an extended-release medicine, when translated to human care,

could prove effective in improving regimen adherence and limiting viral transmission.

## Materials and methods

**Reagents**. DTG was purchased from BOC Sciences (Shirley, NY, USA). Pyridine, dimethylformamide (DMF), $N,N$-diisopropylethylamine (DIEA), myristoyl chloride, stearoyl chloride, behenic acid, octadecanedioic acid, Pluronic F127 (poloxamer 407; P407), polysorbate 20, polyethylene glycol$_{3350}$ (PEG$_{3350}$), ciprofloxacin, 3-(4,5-dimethylthiazol-2-yl)-2,5-diphenyltetrazolium bromide (MTT), dimethyl sulfoxide (DMSO), 1-octanol, paraformaldehyde (PFA), and 3,3′-diaminobenzidine (DAB) were purchased from Sigma-Aldrich (St. Louis, MO, USA). Diethyl ether, ethyl acetate, hexanes, dichloromethylene (DCM), acetonitrile (ACN), methanol, optima-grade water, Dulbecco's Modified Eagle's Medium (DMEM), phosphate-buffered saline (PBS), gentamicin, L-glutamine, potassium phosphate monobasic (KH$_2$PO$_4$), bovine serum albumin (BSA), and Triton X-100 were purchased from Thermo Fisher Scientific/Gibco (Waltham, MA, USA). Cell culture grade water (endotoxin-free) was purchased from Cytiva (Logan, UT, USA). Monoclonal mouse anti-human HIV-1p24 (IgG$_{2a}$; clone 05-001; 1:100 dilution) was purchased from Santa Cruz Biotechnology (Dallas, TX, USA). Polymer-based HRP-

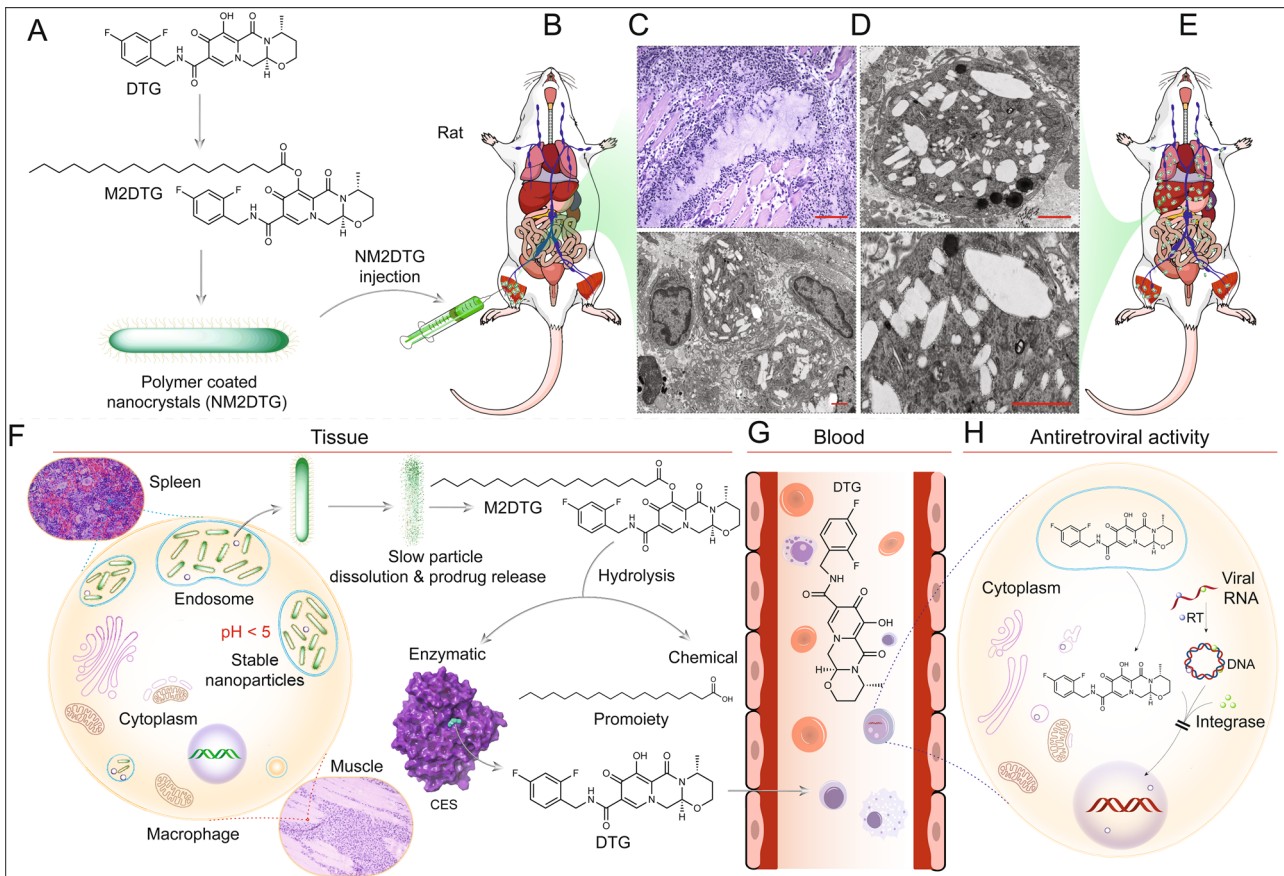

**Fig. 8 Illustration of NM2DTG extended PK profile.** The illustration shows the sequential steps determining the in vivo fate of the NM2DTG nanoformulation. **A** DTG was first esterified with an 18-carbon fatty acid to produce M2DTG which was then nanoformulated to yield NM2DTG. **B** Intramuscular injection of NM2DTG illustrates the formation of the primary drug depot from which the prodrug dissolves from the nanoformulation. **C**, **D** Histiocytic infiltration ensues at the injection site leading to NM2DTG uptake by macrophages. $N = 3$ biologically independent animals/group with representative images shown. Scale bars—400 μm (**C**, upper), 2 μm (**C**, lower; **D**, upper and lower). **E** Biodistribution of drugs to HIV-1 target organs among other tissue sites leads to sustained drug levels in liver, spleen, lymph node, and lung end organs. **F** M2DTG is slowly dissolved from the nanoformulation in the low pH microenvironment in macrophages and then hydrolyzed to release DTG. **G** The slow rate of dissolution of the DTG nanocrystals from tissues and rapid hydrolysis of M2DTG prodrug in plasma allow active DTG to enter the bloodstream and subsequent antiretroviral activities. The two-stage process of dissolution and hydrolysis leads to plasma DTG concentrations at or above the PA-$IC_{90}$, for up to 1 year. **H** DTG shows potent integrase strand inhibition to the integration of viral DNA into the host genome.

conjugated anti-mouse EnVision+ secondary (ref K4001; lot 10137956; no dilution, used as is) was purchased from Agilent Technologies (Santa Clara, CA, USA). Heat-inactivated pooled human serum was purchased from Innovative Biologics (Herndon, VA, USA).

**Synthesis and characterization of DTG prodrugs.** Three monoester prodrugs were synthesized by esterifying the DTG hydroxyl group, yielding lipophilic prodrugs with 14-, 18-, or 22-carbon chains. They are named MDTG[36], M2DTG, and M3DTG, respectively. Additionally, a fourth prodrug M4DTG was synthesized by parallel esterification with two DTG molecules on either end of a single 18-carbon chain. For synthesis, DTG was dried from anhydrous pyridine and then suspended in anhydrous DMF. The mixture was cooled to 0 °C under argon. DIEA (2 equivalents) deprotonated the hydroxyl group of DTG, which was then reacted with 2 equivalents myristoyl- or stearoyl-chloride for 18 hours to create MDTG or M2DTG. M3DTG and M4DTG were prepared by first activating behenic or octadecanedioic acid to their acyl chloride forms using thionyl chloride. The formed chlorides were then reacted with deprotonated DTG enabling the creation of the final prodrugs. These were purified by silica gel column chromatography employing an eluent of 4:1 and then a 9:1 mixture of ethyl acetate and hexanes. The desired compound fractions were acquired from the columns, then dried on a rotary evaporator, precipitated from diethyl ether, and dried from DCM. Finally, the prodrug powders were further dried under a high vacuum providing average chemical yields of 85–95%. Prodrug synthesis was confirmed by proton and carbon nuclear magnetic resonance ($^1$H and $^{13}$C NMR) spectroscopy on a Bruker Avance-III HD (Billerica, MA, USA) operating at 500 MHz, a magnetic field strength of 11.7 T.

**Solubility.** Solubility was determined by adding an excess of the drug to water or 1-octanol at room temperature and mixing for 24 h. Samples were centrifuged at $20,000 \times g$ for 10 min to pellet insoluble drug. Aqueous supernatants were frozen, lyophilized then resuspended in methanol. 1-octanol products were prepared for analysis by dilution in methanol, and samples were analyzed for drug content by UPLC-TUV.

**Nanoparticle preparation and characterization.** For preliminary studies, in vitro works, and studies in mice, nanoformulations of DTG (NDTG) and prodrugs (NMDTG[36], NM2DTG, NM3DTG, and NM4DTG) were manufactured by high-pressure homogenization using P407 as the surfactant. Each solid drug or prodrug was dispersed in a P407 solution in endotoxin-free water to form a presuspension. The drug or prodrug to surfactant ratio was maintained at 10:1 (w/w), and a suspension concentration was in the range of 1–7% (w/v) of drug/prodrug and 0.1–0.7% (w/v) of P407. For studies in rats, formulations were prepared in Phosphate Buffered Saline (11.9 mM potassium phosphate monobasic, 137 mM sodium chloride, 2.7 mM potassium chloride), pH 7.0 at a ratio of 10:0.5:1 (w/w) drug/prodrug:P407:PEG$_{3350}$ and starting drug concentrations between 7 and 11% (w/v). For studies in rhesus macaques, formulations were prepared in the previously mentioned PBS buffer, pH 7.0 at a ratio of 10:1:1 (w/w) prodrug:P407:PEG$_{3350}$ and starting drug concentration of 35% (w/v). To test high concentration formulations for potential clinical translation, formulations were prepared in the previously mentioned PBS buffer, pH 7.0 at a ratio of 7:1:1 (w/w) prodrug:PEG$_{3350}$:Polysorbate 20 and starting drug concentrations between 38-45% (w/v). The presuspensions were homogenized on an Avestin EmulsiFlex-C3 high-pressure homogenizer (Ottawa, ON, Canada) at $20,000 \pm 1000$ PSI to form the desired particle size. Nanoparticles were characterized for hydrodynamic particle diameter (size), polydispersity indices (PDI), and zeta potential as

measured by dynamic light scattering (DLS) using a Malvern Zetasizer Nano-ZS (Worcestershire, UK). The physical and chemical stabilities of the nanoformulations were monitored at 4, 22, and 37 °C. Drug and prodrug concentrations were determined by dissolving the nanoformulations in methanol (1000- to 100,000-fold dilutions). These were then analyzed by UPLC-TUV and processed for sterile use. Endotoxin concentrations were determined using a Charles River Endosafe nexgen-PTS system (Charles River, USA), and only formulations with endotoxin levels <5 EU/kg were used for animal studies. All formulations were suitably syringable and non-viscous enough to pass through a 28 G needle.

**Isolation and cultivation of human monocyte-derived macrophages (MDM)**. Human monocytes were obtained by leukapheresis from HIV-1/2 and hepatitis B seronegative donors and purified by counter-current centrifugal elutriation. Monocytes were cultured in conditions detailed in our previous works[29,36]. After differentiation, MDM were used for the drug-particle uptake, retention, and antiretroviral assays.

**Drug nanoparticle uptake and retention**. The in vitro assessment of nanoformulation uptake and retention in MDM were performed as detailed in our previous works[29,36]. For drug nanoparticle cellular uptake studies, MDM were treated with 5 or 25 μM NDTG, NMDTG, or NM2DTG and collected at 2, 4, 8, and 24 h following treatment. For retention studies, MDM was treated with 5 or 25 μM NDTG, NMDTG, or NM2DTG for 8 h, and washed cells were collected at days 1, 10, 20, and 30, to be analyzed for drug and prodrug content by UPLC-TUV.

**Morphological evaluation of intracellular nanoparticles**. MDM were treated with 25 μM NM2DTG for 8 h, and collected at days 0, 10, 20, and 30 after treatment and analyzed by transmission electron microscopy (TEM) to visualize intracellular nanoparticles as detailed in our previous works[29,36]. Images were acquired digitally with an AMT digital imaging system (Woburn, MA, USA).

**Measurements of antiretroviral activities in HIV-1-challenged MDM**. The study comprised of MDM treatments with 1 or 10 μM NDTG or NM2DTG for 8 h. At 5, 10, 15, 20, 25, and 30 days after treatment, the cells were challenged with HIV-1$_{ADA}$ at a multiplicity of infection (MOI) of 0.1 infectious particles/cell for 4 h according to previously established protocols[29,36,40]. Cells were fixed in 4% PFA at each time point, and expression of HIV-1p24 antigen was determined by immunocytochemistry.

**Measures of the half-maximal inhibitory concentration (IC$_{50}$) of DTG formulations**. The IC$_{50}$ determination in MDM was conducted as per previously established protocols[29,36], wherein the cells were treated with a range of drug concentrations, 0.01–1000 nM of DTG, MDTG, or M2DTG dissolved in 0.1% (v/v) DMSO for 1 h prior to challenge with HIV-1$_{ADA}$ (MOI of 0.1) for 4 h. Cell supernatants were collected on day 10 and assayed for HIV-1 RT activity.

**PK studies in rodents and rhesus macaques (RM)**. Male Balb/cJ mice (23–26 g, 6–8 weeks, Jackson Labs, Bar Harbor, ME, USA) were administered a single intramuscular (IM; caudal thigh muscle) of 45 mg DTG-equivalents (eq.)/kg of NDTG, NMDTG, NM2DTG, NM3DTG, or NM4DTG in a maximum volume of 40 μL/25 g mouse using a 28 G × ½" needle[36]. Animals were housed under a 12-h light/dark cycle at a temperature of 20–24 °C and a humidity range of 30–70%. The animals were maintained on a sterilized 7012 Teklad diet (Harlan, Madison, WI), and acidified water was provided ad libitum. Following injection, blood samples were collected into heparinized tubes on day 1 post-drug administration and then weekly until 1 year by cheek puncture (submandibular vein) using a 5 mm lancet (MEDIpoint, Mineola, NY, USA). Blood samples were centrifuged at 2000 × $g$ for 8 min for plasma collection and drug content quantitation. On day 367, after drug administration, animals were humanely euthanized using isoflurane followed by cervical dislocation.

Male Sprague-Dawley (SD) rats (186-225 g, 8 weeks, SASCO, Wilmington, MA, USA) were administered a single intramuscular (IM) dose in the caudal thigh muscle of 45 mg DTG-eq./kg of NDTG, NM2DTG, or sterile saline in a maximum volume of 200 μL/200 g rat using a 28 G × ½" needle. Animals were housed under a 12-h light/dark cycle at a temperature of 20–24 °C and a humidity range of 30–70%. The animals were maintained on a sterilized 7012 Teklad diet (Harlan, Madison, WI), and acidified water was provided ad libitum. Following injection, blood samples were collected in heparinized tubes at 4 h, day 1, 4, and 7 after administration, then weekly for 6 months and monthly from time points of 6 months to 1 year. Blood draws were made through retro-orbital plexus bleeds. Blood samples were centrifuged at 2000 × $g$ for 8 minutes for plasma collection and quantitation of plasma drug contents. At days 57, 175, and 364 following drug administrations, animals were humanely euthanized, and tissues (spleen, liver, lymph nodes [pooled; cervical, axillary, and inguinal], muscle [site of injection], kidney, lung, gut, brain, and rectal tissue) were collected for quantitation of DTG and prodrug levels and/or histology. Part of each tissue was placed in an Eppendorf tube on dry ice and stored at −80 °C for later drug analysis. Each tissue was placed in 10% neutral buffered formalin for immunohistochemistry, pathology, and toxicity studies. On day 3, following drug administration, animals were humanely

euthanized, and muscle tissue from the injection site and contralateral control muscle were collected for drug analysis, immunohistochemistry, and pathologic and electron microscopy studies. The muscle tissue containing the site of injection was collected, divided in half, and fixed appropriately for either immunohistochemistry/pathology or electron microscopy. For histological examination, 5 μm sections of paraffin-embedded tissues were stained with hematoxylin and eosin (H&E). Images were captured using a Nuance EX multispectral imaging system affixed to a Nikon Eclipse E800 microscope (Nikon Instruments, Melville, NY, USA). A board-certified pathologist conducted a histopathological assessment according to the Society of Toxicologic Pathology[50]. Toxicity in SD rats was assessed by evaluating complete blood counts, serum chemistry profiles, and histological examination[39]. At sacrifice time points, blood was collected into potassium-EDTA coated tubes for hematology analysis using a VetScan HM5 veterinary hematology blood analyzer (Abaxis Veterinary Diagnostics, Union City, CA, USA). Serum chemistry profiles were determined using a VetScan comprehensive diagnostic profile disc and a VetScan VS-2 instrument (Abaxis). Results for treated animals were compared to those from age-matched untreated control rats.

Female rhesus macaques (RM, *Macaca mulatta*; 5.5–7.5 kg, 8–13 years old, New Iberia Research Center, New Iberia, LA, USA) were administered an intramuscular (IM) dose in the quadriceps muscle of 45 mg DTG-eq./kg NM2DTG in a maximum volume of 0.5 mL/kg using a 23 G × 1½" needle (not to exceed 1.5 mL/injection site; opposite quadriceps were used if the volume required multiple injections). The NM2DTG nanoformulations were prepared in the Nebraska Nanomedicine Production Plant by established good laboratory practice (GLP) protocols[31]. Animals were housed under a 12-h light/dark cycle at a temperature of 20–24 °C, humidity range of 30–70%, and television for entertainment throughout the experimental duration. The animals were fed daily on 5045 Purina monkey diet (Neenah, WI, USA) supplemented with fresh fruit or vegetables, and water was provided ad libitum. These animals were previously exposed to SHIV or Zika virus and remained uninfected and transferred to these studies. All the animals were observed daily by animal care personnel/veterinary staff of Comparative Medicine at the University of Nebraska Medical Center (UNMC). Following injection, blood samples were collected into EDTA tubes at days 1, 3, 7, and 14 after administration, then biweekly until 3.5 months and monthly from 3.5 months to 6.5 months. Blood draws were made following ketamine anesthesia. Blood samples were centrifuged at 1000 × $g$ for 20 min to collect and quantify plasma drug contents. General animal well-being and recorded movement and skin reactions were recorded. A second booster dose was given on day 217 in the same manner. Blood samples were collected on days 1, 3, 7, 14, and 21 after a boost, then biweekly until 11.25 months (total) and monthly until day 428. At days 393, 400, and 428 following initial drug administration, animals were euthanized, and tissues were collected for quantitation of DTG and prodrug levels and/or histology. Part of each tissue was placed in an Eppendorf tube on dry ice and stored at −80 °C for later drug analysis. Part of each tissue was also placed in 10% neutral buffered formalin for immunohistochemistry, pathology, and toxicity studies. The muscle tissue containing the site of injection was collected, divided, and fixed appropriately for either immunohistochemistry/pathology or electron microscopy. For histological examination, 5 μm sections of paraffin-embedded tissues were stained with hematoxylin and eosin (H&E). Images were captured using a Nuance EX multispectral imaging system affixed to a Nikon Eclipse E800 microscope (Nikon Instruments, Melville, NY, USA). A board-certified pathologist conducted a histopathological assessment according to the Society of Toxicologic Pathology[50].

The rodent sex choices were based on the potential that female hormones during estrous cycles could cloud registered PK data sets and elicit data variability[51]. The female NHPs served to confirm the rodent data sets.

DTG and M2DTG were quantitated in mouse, rat, and rhesus plasma and tissues by UPLC-tandem mass spectroscopy (MS/MS) using a Waters ACQUITY H-class UPLC connected to a Xevo TQ-S micro mass spectrometer and described in the supplementary methods. All solvents for sample processing and UPLC-MS/MS analysis were Optima-grade (Fisher). Non-compartmental PK for plasma DTG in all species was performed with Phoenix WinNonlin-8.3.3.33 software (Certara, Princeton, NJ, USA).

**Transmission electron microscopy (TEM)**. TEM tissue and cell samples were processed according to previously optimized protocols[29,38] and examined on a Tecnai G$^2$ Spirit *TWIN* (Thermo Fisher Scientific) operating at 80 kV.

**Prodrug hydrolysis kinetics in tissue and plasma**. Male SD rats (SASCO) were humanely euthanized, perfused, and tissues collected as described previously. Samples were stored at −80 °C until further processed as described previously. M2DTG prodrug solution, dissolved in 1% (v/v) methanol and aqueous prodrug nanoformulation (NM2DTG) were used as substrates for cleavage in various rat tissue homogenates. After preincubation of 100 μL of tissue homogenate at 37 °C for 5 min, the reactions were initiated by the addition of the substrates and stopped by the addition of 900 μL of acidified methanol (0.1% formic acid and 2.5 mM ammonium formate in Optima-grade methanol) at 30 min, 2, 6, 12, 24, and 48 h time points. Control samples were incubated using the same method but with substrates added after adding acidified methanol. The mixtures were centrifuged at 16,000 × $g$ for 10 min to remove precipitated protein. The supernatants were

aspirated and stored at −80 °C until analysis. The supernatant was diluted 1:1 with internal standard (IS: DTG-d3 20 ng/mL, 40 ng/mL SDRV, 40 ng MDRV), vortexed for 30 s, and transferred to a 96-well plate to be injected onto the UPLC-MS/MS system for drug quantitation and described in the Supplementary Methods.

**pH affects for prodrug cleavage**. The contribution of pH to prodrug cleavage was studied in pH buffers including 7.5 mM ammonium acetate (pH 6.0, adjusted with acetic acid), 7.5 mM ammonium bicarbonate (pH 8.0, adjusted with acetic acid), and pH adjusted solutions including 0.1% formic acid (pH 2.0), and 0.1% ammonium hydroxide (pH 10.3). M2DTG prodrug solution and NM2DTG suspension were incubated in these matrices as per the protocol for rat tissue homogenates and analyzed as such.

**Statistics and reproducibility**. For all studies, data were analyzed using Microsoft Excel V16.49 (Redmond, WA, USA) and GraphPad Prism 9.3 software (La Jolla, CA, USA) and presented as the mean ± the standard error of the mean (SEM). Exclusion criteria were predetermined. Extreme outliers beyond the 99% confidence interval of the mean and 3-fold greater than the SEM were excluded. Excluded values are highlighted in blue with an asterisk in the Source data files. Significant differences were determined at $P < 0.05$.

Sample sizes chosen were sufficient to determine significance in all assays, with reproducible statistically significant differences between experimental conditions. Experiments were performed using a minimum of three biologically distinct replicates. Samples sizes were not based on power analyses. For comparing two groups for PK analysis, six animals/group will provide 80% power at the 0.05 level of significance to detect a difference of 2.0 standard deviations using a $t$ test. For animal studies, sample sizes were determined to provide statistical power while also meeting cost and ethical criteria for animal use. For all studies, samples/cells/animals were randomly allocated into experimental groups at the beginning of each study.

All attempts to reproduce the experimental findings were successful. For chemical synthesis, characterization, and formulation production, experiments were repeated independently a minimum of three times with similar results. For in vitro cellular assays, experiments were repeated independently a minimum of three times with similar results. For prodrug hydrolysis studies, experiments were repeated independently two times with equivalent results. For animal studies, the results of the yearlong study were validated in three different species (mice, rats, and rhesus macaques). Mouse studies were conducted once with a starting $N = 5$ animals per group, and rat experiments were conducted once with a starting $N = 13$ animals per group (subdivided into three different sacrifice time points where $N = 3$ for 2 months, $N = 4$ for 6 months, and $N = 6$ for 1 year), and rhesus macaque studies were conducted once with an $N = 3$ animals.

Due to experimental, limitations investigators were not blinded in conducting experiments or sample collection; instead relying on an unbiased approach. However, attempts were made to generate unbiased data through blinded data collection/analysis. For example, separate investigators conducted sample collection and data collection/analysis for all animal experiments (drug level determination, CBC counts, serum chemistry analysis), providing support to the unbiased conduct of the data generated. For pathological evaluation of histology sections of tissues, the pathologist was blinded.

**Study approvals**. All experimental protocols involving the use of laboratory animals were approved by the University of Nebraska Medical Center (UNMC) Institutional Animal Care and Use Committee (IACUC) in accordance with the standards incorporated in the Guide for the Care and Use of Laboratory Animals (National Research Council of the National Academies, 2011) ensuring the ethical care and use of laboratory animals in experimental research. All animal studies were performed according to UNMC institutional and National Institutes of Health (NIH) guidelines for laboratory animal housing and care in American Animal Association and Laboratory Animal Care (AAALAC) accredited facilities. Human peripheral blood monocytes were isolated by leukapheresis from HIV-1/2 and hepatitis B seronegative donors and purified by centrifugal elutriation from the UNMC Elutriation and Cell Separation Core according to a UNMC Institutional Review Board (IRB) exempt protocol with informed consent.

**Reporting summary**. Further information on research design is available in the Nature Research Reporting Summary linked to this article.

## Data availability

Source data are provided in the Figshare database under Digital Object Identifier https://doi.org/10.6084/m9.figshare.19026452 (Fig. 1), https://doi.org/10.6084/m9.figshare.19026983 (Fig. 2), https://doi.org/10.6084/m9.figshare.19027175 (Fig 3), https://doi.org/10.6084/m9.figshare.19027229 (Fig. 4), https://doi.org/10.6084/m9.figshare.19027253 (Fig. 5), https://doi.org/10.6084/m9.figshare.19362500 (Fig. 6) and https://doi.org/10.6084/m9.figshare.19027397 (Supplementary Materials). The three-dimensional crystal structures of CES1 (PDB ID: 1YA8; PDB DOI: 10.2210/pdb1YA8/pdb) protein were retrieved from the Research Collaboratory for Structural Bioinformatics Protein Data Bank (RCBS PDB) database.

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

## Acknowledgements

We wish to thank the University of Nebraska Medical Center Cores: Electron Microscopy (Tom Bargar, Nicholas Conoan, and Susan Brusnahan), Elutriation and Cell Separation (Myhanh Che and Na Ly), Tissue Sciences Facility (Melissa Holzapfel), and Comparative Medicine for technical assistance. We thank Dr. Shah Valloppilly of the University of Nebraska-Lincoln Nebraska Center for Materials and Nanoscience X-Ray Structural Characterization Facility for support in characterizing the modified antiretroviral drugs used in this study. We also thank Mohammad Ullah Nayan and Srijanee Das for animal support. Finally, Dr. Prasanta K. Dash, Emiko Waight, Edward Makarov, and Kabita Pandey for technical assistance for tissue sectioning for histopathology. Funding was provided by. National Institutes of Health grants R01 NS034239; T32 NS105594; R01 NS036126; R01 MH115860; and R01 NS126089 (HEG). National Institutes of Health grant R01 AI145542 (BE and HEG). National Institutes of Health grant R01 AI158160 (HEG and BE). National Institutes of Health grant R01 MH121402 (HEG and BDK). National Institute of Health grant R01 AI129745 (SNB). University of Nebraska Foundation (donations from the Carol Swarts, M.D. Emerging Neuroscience Research Laboratory; the Margaret R. Larson Professorship; and the Frances and Louie Blumkin, and Harriet Singer Endowments). Vice Chancellor for Research Office at the University of Nebraska Medical Center core facilities.

## Author contributions

Conceptualization: S.D., B.S., B.E., H.E.G. Methodology: S.D., B.S., S.N.A., A.T.P., J.M.M., N.G., A.S., S.N.B., A.K.D., A.D., S.N.B., Y.A., A.Y., B.E., H.E.G. Investigation: S.D., B.S., A.N.B., S.N.A., A.T.P., B.H., B.L.D.S., A.S., M.J., M.T., D.J.M., M.M., A.Y., S.M.C., B.E. Visualization: S.D., B.S., M.M., B.D.K. Funding acquisition: B.E., H.E.G. Project administration: B.S., J.M.M., A.S., M.J., S.N.B., A.K.D., A.D., Y.A., B.E., H.E.G. Supervision: B.S., A.N.B., J.M.M., S.N.B., A.K.D., A.D., Y.A., B.E., H.E.G. Writing manuscript: S.D., B.S., H.E.G. Writing—review and editing: S.D., B.S., S.N.A., A.T.P., A.Y., B.E., H.E.G.

## Competing interests

B.E. and H.E.G. are named inventors on patents (US011166957B2, WO2021035114A1, EP3886863A1, WO2019199756A1, US20200390796A1) that cover the medicinal and polymer chemistry technologies employed in this manuscript, encompassing the synthesis of long-acting dolutegravir prodrug and formulation manufacturing. A.Y., B.E., and H.E.G. are co-founders of Exavir Therapeutics, Inc., a biotechnology company that is developing extended-release long-acting antiretroviral drugs. H.E.G. is the Interim Director and B.S. is the Operations Manager of the Nebraska Nanomedicine Production Plant, a good manufacturing program facility. The authors declare that this work was produced solely by the authors and that no other individuals or entities influenced any aspects of the work including, but not limited to, the study conception and design, data acquisition, analyses and interpretation, and writing of the manuscript. No other entities provided funds for the work. The authors further declare that they have received no financial compensation from any other third parties for any aspects of the published work. The remaining authors declare no competing interests.
