## [Peer Review File · Nature Communications]

REVIEWER COMMENTS

Reviewer #1 (Remarks to the Author):

In the manuscript "Pathways for Dolutegravir Transformation from a Daily Oral to a Once-a-Year Parenteral Medicine" authors described new prodrugs of DTG, integrase HIV inhibitor, and its nanoformulation. DTG was modified by esterification of the DTG hydroxyl group with 14-, 18-, or 22-carbon chains, and by the esterification of two DTG molecules on either end of a single 18-carbon chain. The 14-carbon chain DTG prodrug was characterized in a previous publication (reference 29). The 18-carbon chain DTG prodrug (denoted M2DTG) showed the best PK profile and was further characterized. LA formulations of ARV are promising approach to improve adherence to medication.

Comments:

1. PK analysis was done in 3 species with the fixed dose of 45 mg/kg. However, target for those studies is not clear. Currently, DTG is use in combination with other antiretrovirals at 50 mg once daily for HIV treatment. This daily dose results in plasma C_{max} ~3ug/ml and C_{trough} ~0.8ug/ml. The dosing of nanoformulated M2DTG reach this level only for a couple days in some species, not a year. With no efficacy studies, it is difficult to assess if the nanoformulation will deliver sufficient amount of drug. Based on data provided here, it seems that the dose will need to be increased at least 13 times for mice and rats to reach therapeutic level for one year. Is this feasible with the type of formulation and it is possible to translate to human? Please, specify and justify the concentration target for the PK studies.
2. Will the increased dose of the nanoformulation in this technology result in the increase of drug in plasma or will it extend the delivery time?
3. PK studies are using only one sex (male mice and rats, female NHP). Is there any reason for this? Please, justify in the method section.
4. Following statement: "The prodrug presence at the injection site allows a convenient drug removal if and when warranted for secondary adverse reactions" is not supported with any data and should be removed. Intramuscular idepots of nano formulations are very difficult to remove without muscular damage. In addition, this statement contradicts several results of the study: Figure 2B shows long retention in macrophages (30 days) that would not be affected by the removal of the depot from injection site. Figure 6 d, e, f shows nanocrystals in endosomes of multiple organs, not at the injection site.

5. Fig 2 has 1 μm and 10 μm instead 1 μM and 10 μM

6. In Figure 2 D-O, please identify nanocrystals and endosomes for clarity as in Supp. Figure 12.

Reviewer #2 (Remarks to the Author):

Deodhar et al. created and screened a library of monomeric and dimeric dolutegravir (DTG) prodrug nanoformulations. They identified a prodrug nanocrystal formulation (NM2DTG) that exhibited a far slower decay rate than DTG and maintained levels above 1xPAIC90 for up to a year in mice and rats. The authors show that the muscle injection site and lymphoid tissues were depots of prodrug hydrolysis and characterize various factors that affect decay rate. Overall, the data on NM2DTG look very promising and support its ultralong acting profile. The authors indicate PrEP as a main use for NM2DTG as it provides a much longer dosing period than the recently approved bimonthly long acting cabotegravir (CAB LA). This work is similar to a recent report by the same group on a CAB prodrug nanocrystal formulation.

I have the following comments:

1- As the authors acknowledge, the CAB LA trial results demonstrate that plasma concentrations equivalent to 4xPAIC90 are a target benchmark for PrEP protection. Given the high similarities between CAB and DTG, the authors may want to explicitly indicate this benchmark as a target profile instead of "exceeding the 1xPAIC90 for one year" stated in the Introduction. Certainly, the authors can adjust this benchmark if they generate data from challenge studies in macaques supporting a lower threshold for complete protection. However, in the absence of such data the 4xPAIC90 in macaques should remain the working target level. Because the macaque model has now been found to be predictive of clinical efficacy, data from macaques receive heightened attention.

2- While the authors are to be complimented for the comprehensive characterization of NM2DTG, the PK data in macaques dosed with 45 mg DTG-eq./kg showing plasma concentrations falling rapidly under 4xPAIC90 after the initial and the booster dose on day 217 are a bit underwhelming. The authors do not show any data from dose escalation studies in macaques that demonstrate maintenance of DTG above 4xPAIC90 for an extended period of time that approaches a year and would translate to a year in humans. The authors singled out rat data for dose extrapolation to humans which may be less convincing than macaque data.

Other comments:

- Please add needle size used for the IM injection, and comment on viscosity of the formulation
- Line 262: clarify if the IM injection was done in the same or opposite quadricep
- Line 396: Clarify the H&E was done in rats
- Figure 3C is missing the dotted line for 4xPAIC90
- Line 441: are drug levels in all lymph nodes similar (mesenteric vs axillary or inguinal? Do the nodes closer to the injection site have higher drug exposures?
- Line 450: describe what modest metabolic differences were observed

Reviewer #3 (Remarks to the Author):

The study in the manuscript, "Pathways for Dolutegravir Transformation from a Daily Oral to a Once-a-Year Parenteral Medicine" is among the most important finding, to date, that describes a pathway to effectively treat HIV. The information presented here is built upon the notion that although ART drugs are available for all the HIV-infected individuals, the treatment is not very effective for a variety of reasons. For examples, lack of medication adherence, toxicity, drug interactions, social stigma, prevalence of drugs of abuse, and behavior. However, if the burdens of the ART drug regimen is reduced from daily or weekly to monthly or even yearly, then the treatment could be highly effective in HIV populations. Over the years, the scientific community has been working on finding a long-acting ART drug for HIV treatment. In this respect, the finding here strongly provides the evidence that dolutegravir, an integrase inhibitor, can be used as intramuscular depot that ultra-slowly releases the drugs up to a year.

In this context, the study is highly significance in the HIV field. The work is also original and results clearly support the conclusions. This will significantly advance the field.

The methodologies in this manuscript is also robust, because the study not only synthesized nano-prodrugs and extensively characterized them, but also used in-vitro primary macrophages and three appropriate animal models, mice, rat, and rhesus to characterize the drug formulations. They are also described well.

Although the number of biological replicates came from fewer cells or animals (mostly 3-4), the data from the final formulation NM2DTG, compared to others, are so significant and noticeable that, additional biological replicates would not offer any additional power or significance.

The manuscript is well-written in a lay-men language and builds an impressive story of its future application upon clinical studies in humans.

The overall data, both in manuscript main figures and supplemental figures are all clean, well-presented, and comprehensive. The statistical analyses are also appropriate.

The reviewer found a very few minor errors that need to be dealt with before its acceptance.

1. There is no explanation, if relevant, why rats were male and rhesus were female. Both male and female are important to be considered in the study, unless its justified. Its understood that with 3-4 animals, its hard to chose equally both the sexes.
2. In results section, fig 1c does not have PK profile, however, its mentioned in the text, "drugs affect both the PK and aqueous solubility".
3. In results section, the description of fig 3c comes after fig 3d-g. Its important to describe the figure in the text in the same order that they are presented in the figure.
4. Figure 6 is an important summary figure. However, its presentation appears to be missing in the discussion section.

March 29, 2022

Re: NCOMMS-22-01074 "Transformation of Dolutegravir to an Ultra-Long-Acting Parenteral Prodrug Formulation"

Thank you for the thorough and reasoned review of our manuscript. We are pleased to respond to each of the queries raised point-by-point. The amendments made in the text **are highlighted in yellow.**

Reviewer 1

Overview. *"In the manuscript "Pathways for Dolutegravir Transformation from a Daily Oral to a Once-a-Year Parenteral Medicine" authors described new prodrugs of DTG, integrase HIV inhibitor, and its nanoformulation. DTG was modified by esterification of the DTG hydroxyl group with 14-, 18-, or 22-carbon chains, and by the esterification of two DTG molecules on either end of a single 18-carbon chain. The 14-carbon chain DTG prodrug was characterized in a previous publication (reference 29). The 18-carbon chain DTG prodrug (denoted M2DTG) showed the best PK profile and was further characterized. LA formulations of ARV are promising approach to improve adherence to medication."*

Response: We are pleased to address the queries raised for once-a-year dosing. *First*, new experiments were performed in nonhuman primates (NHPs) who were given an initial and boosting doses with a one year follow on. In these experiments tissue drug levels were monitored. The normal tissue histology examined by an experienced anatomical pathologist highlight early drug safety. The data provides support to a 6 to 12 month NM2DTG formulation treatment. *Second*, specific extensions of the data sets are provided. Notably, pharmacokinetic (PK) modeling and simulations were explored to predict the human dosing required to achieve protective plasma drug levels of 4x PA-IC₉₀ at C-min. The simulation work is based on extrapolation of animal and published DTG PK data that 4x PA-IC₉₀ target plasma drug levels can be achieved throughout a once yearly dose at achievable injection volumes. These are also in line with current US Food and Drug Administration (FDA)-approved long-acting injectables that are common in clinical practice. *Third*, tissue NHP distribution and drug tissue data sets in NHP which were added in the text including lymphoid tissue and muscle injection site drug levels are provided in the **new figures 4 and 7**. *Fourth*, the added data sets show the injection site nanocrystal drug depot within histocytes "macrophages" with extended infiltration at the muscle injection site. Drug values were supported by modeling. This affirms that protective DTG plasma concentrations can be reached in humans at achievable injection volumes. *Also*, we highlight the preclinical studies currently underway by Exavir Therapeutics (<https://exavirtherapeutics.com>). These serve to support an Investigational New Drug application submission to the US FDA and other regulatory agencies worldwide for NM2DTG use in human trials. title was amended to "ultra-long-acting" (XLA) dosing. As we have not yet administered the formulation to humans the focus is from 6 to 12 months administration. These more conservative simulations more precisely reflect the extension of NM2DTG's apparent half-life.

Point 1. *"PK analysis was done in 3 species with the fixed dose of 45 mg/kg. However, target for those studies is not clear. Currently, DTG is use in combination with other antiretrovirals at 50 mg once daily for HIV treatment. This daily dose results in plasma C_{max} ~3 ug/mL and C_{trough} ~0.8 ug/mL. The dosing of nanoformulated M2DTG reach this level only for a couple days in some species, not a year. With no efficacy studies, it is difficult to assess if the nanoformulation will deliver sufficient amount of drug. Based on data provided here, it seems that the dose will need to be increased at least 13 times for mice and rats to reach therapeutic level for one year. Is this feasible with the type of formulation and it is possible to translate to human? Please, specify and justify the concentration target for the PK studies."*

Response: The current study is unique for several reasons. *First*, is regarding rigor and reproducibility. The current report builds on several of our prior works (Nature Materials 2020 19(8) pages 910-920). *Second*, rigor and reproducibility were built on our prior validations with contract laboratories (Covance Laboratories, a global contract research organization and drug development services company). *Third*, the analytical analyses were repeated from samples of plasma and tissue in three species. We call attention to the prior studies demonstrating plasma levels above 4x the PA-IC₉₀ relevant to the integrase inhibitor cabotegravir (CAB) as is required for simian immunodeficiency virus (SIV) prevention. The works, including what was reported in AIDS 2017 31(4):461-467, demonstrate that the terminal phase half-life of ARVs in macaques is shorter (3-12 days) when compared to humans (21-50 days). The required drug administration was sequential dosing at 50 mg/kg CAB in NHPs for pre-exposure prophylaxis (PrEP). Notably, a human equivalent dose of 50 mg/kg in NHPs used for PrEP studies translated to 1129 mg of CAB in humans, and higher than the 400 and 600 mg now used for monthly or bimonthly dosing. *Fourth*, the **NM2DTG dose need consider drug penetrance into lymphoid viral reservoirs. NM2DTG concentrations are a log greater than nanoformulated native DTG.** PK profiles show limited fluctuation with NM2DTG compared to NDTG when employing 400 mg/mL dosing volumes. Future dose escalation and PrEP experimental testing will be required for dose optimizations as performed with the native drug (Br J Clin Pharmacol. 2015 Sep;80(3):502-14) and see below. *Fifth* and most importantly, we also provide new data in modeling and for tissue drug and prodrug analyses to justify feasibility of XLA dosing intervals which will be further interrogated in now a planned phase I human clinical trial. These are projections that will be written up in a subsequent paper after completion. Notably, the minimum required plasma concentration of 4x the PA-IC₉₀ for DTG is 256 ng/mL which makes it possible for NM2DTG formulations to provide effective plasma concentration at human achievable injection doses and volumes. Since the prodrug MDTG accumulates in lymphoid tissue, providing higher DTG than seen by DTG alone supports the notion that effective therapy is possible with plasma concentrations of less than 256 ng/ml. Namely, we point to currently FDA-approved long-acting injectable prodrug formulations which provide a reference point for achievable human doses and injection volumes used in clinical practice. Three methods of PK modeling that suggest NM2DTG can achieve protective target drug concentrations of 4x PA-IC₉₀ when administered in a dosage form and injection volume within the range of the approved products. It is noteworthy that other nanoformulated drugs developed for slow release, well-studied in animals and in humans, provide anchors for dose required injection volumes for human administration. Namely, we refer to both CABENUVA and APRETUDE. The approved nanoformulations of CAB for HIV-1 treatment and prevention are administered as 900 mg 3 ml injections. Moreover, the recently developed esterified antipsychotic prodrugs are a direct analog of our approach. Indeed, esterification is used to produce prodrugs with extended-release rates from injectable nanoformulations. This is typified by the recently approved INVEGA HAFYERA, which is a once-every-six-month injectable of paliperidone palmitate (the ester of paliperidone an atypical antipsychotic, and previously approved as a daily oral agent. INVEGA HAFYERA) is administered as a 1.6g / 5mL intramuscular injection. During NMDTGs development and investigations, PK models have been used to describe the disposition, distribution, elimination, and dosing of M2DTG and DTG following NM2DTG injection. Herein, we provide three methods of PK modeling to support human dose and volume projections within ranges and feasibility of XLA dosing intervals. These include simulations to determine doses required to achieve protective levels in rodents along with body-surface-area scaling factors provided in guidance from US FDA for rodent-to-human dose translation; mechanistic-based PK model (MBPK) in accordance with previously published methods based on our observed NHP data; and incorporation of *in vivo* DTG release kinetics into previously published human population PK models. Although all approaches suggest different human doses and all point to feasible doses and volumes for human administration. *Fifth*, we performed a modeling exercise to estimate the human equivalent dose based on simulations of animal PK data sets and subsequent body-surface-area scaling factors provided in US FDA guidance for dose translation. As seen in the figure, Monte Carlo simulations based on observed data suggest that the vast majority of mice and rats given doses of 175 to 200 mg/kg DTG-eq. would achieve plasma concentrations of 4x PA-IC₉₀ at the 6 month time (**figure 1 and 2 response**). Using the body-surface-area scaling factors provided in published guidance by the US FDA, these doses would translate to roughly 15-35 mg/kg DTG

eq. in a human, which in a 60kg human would translate to 1.5-3.4g of prodrug and 4-9 mL total injection volume and could be given as one or two injections reflective of INVEGA HAFYERA.

Figure 1 response. Probability of attainments (PTA) of 4x PA-IC₉₀ plasma drug concentration by simulated PK evaluations. The evaluations follow a single intramuscular NM2DTG dose in mice (A) and rats (B). Monte Carlo simulations (MCSs) were used to assess estimated DTG plasma concentrations. The data demonstrates timed percentages above each of the increasing DTG PA-IC₉₀ values. The 1x, 2x, 3x, and 4x PA-IC₉₀ DTG concentrations are listed as 0.064, 0.128, 0.193 and 0.256 mcg/mL, respectively. Simulated doses of 85 to 500 mg DTG-eq./kg (in mice; A) and 65 to 300 mg DTG-eq./kg (in rats; B) are shown at 6 months.

Figure 2 response (below). Population PK parameters. Goodness-of-fit plots for Population and Bayesian (“individual”) predicted to observed NM2DTG concentrations (units for both axis in mcg/mL). Population predictions are based on the mean PK values for the population whereas the Bayesian predictions are each animal’s own unique PK profiles.

Mice: Preliminary Population PK Parameters

PK parameter	mean	SD	CV%	Var	median	shrink%*
Ke (h ⁻¹)	0.000238	0.000225	94.27	5.05E-08	0.00014	17.74
Ka (h ⁻¹)	2.46	3.34	135.71	11.16	0.95	17.97
V (L)	4.43	0.35	7.98	0.13	4.2	14.87
Tlag (h)	13.63	1.2	8.85	1.46	14.18	17.97

Abbreviations: Ke: elimination rate constant, Ka: absorption rate constant, V: volume, Tlag: delay variable, SD: standard deviation, CV: coefficient of variation, Var: variance

*ratio of the mean posterior parameter value variance across all subjects to the total population

Observed versus Predicted NM2DTG Concentrations Plots for Mice

Rats: Preliminary Population PK Parameters

PK parameter	mean	SD	CV%	Var	median	shrink%*
Ke (h ⁻¹)	0.000501	0.000119	23.82	1.42E-08	0.000496	19.97
Ka (h ⁻¹)	3.41	1.48	43.24	2.18	2.738	27.53
V (L)	12.21	1.16	9.51	1.35	12.82	14.62
Tlag (h)	14.42	1.99	13.78	3.95	15.24	30.25

Abbreviations: Ke: elimination rate constant, Ka: absorption rate constant, V: volume, Tlag: delay variable, SD: standard deviation, CV: coefficient of variation, Var: variance

*ratio of the mean posterior parameter value variance across all subjects to the total population

Observed versus Predicted NM2DTG Concentrations Plots for Rats
 Population Bayesian ("Individual")

Sixth, a preliminary mechanistic-based PK (MBPK) model was developed to support the feasibility of ultra-long-acting dosing intervals in line with previously published methods that have been validated for other antiretroviral agents (Frontiers in Pharmacology 2020 Dec 18;11:603242). Herein, the multi-phasic absorption of DTG from NM2DTG was modeled by a three-component system based on PK parameters of absorption observed in the NHPs and known prodrug-to-drug hydrolysis rates observed in vitro. The model was validated by accurate prediction of the observed NHP PK (**figure 3 response**). Primate parameters were then scaled to human parameters using allometry and known values for parent DTG provided in the US FDA labeling. The subsequent human model can then be used to predict the potential human PK of NM2DTG, which are provided in **figure 4 response**. This model predicts that 800 mg of DTG-equivalent (or approximately 1.3 g of prodrug) would readily provide coverage above 4X-PAIC90 throughout a 6-month dosing interval. Provided that our current formulation provides 400mg/mL of drug content, a 1.3 g /3mL injection is well within the bounds of currently FDA approved products, such as APRETUDE which is administered as a 900 mg /3 mL injection, or INVEGA HAFYERA which is administered as a 1.6 g /5mL injection.

Figure 3 response. A three-component mechanistic based PK model MBPK was developed based on observed NHP data to model the projected time-course of exposure in humans (left). The model was validated and shown to accurately predict the observed non-human primate data for plasma M2DTG and DTG exposure (right).

Figure 4 response. Solid lines show the projected human plasma PK for M2DTG and DTG based on the MBPK model scaling to human parameters following an 800mg DTG-eq. dose. This dose was estimated to provide coverage above the protective 4C-PAIC90 level for over 180 days with a single injection (left), which could be further extended through a booster dose that would benefit from drug accumulation (right).

In a *third* modeling approach, we explored potential human NM2DTG dosing regimens by human population PK modeling. By this method, DTG release kinetics as noted following NM2DTG in our non-human primate studies were incorporated into a published population PK human model (Br J Clin Pharmacol. 2015 Sep;80(3):502-14). As noted in our studies, overall DTG release rates were similar in non-human primates and rats. Such modeling allowed the evaluation of not only dose and interval but also the influence of inter-subject variability in DTG PK on expected DTG plasma concentration profiles. Modeling emphasized the range of DTG plasma concentrations at the end of dosing intervals and not the transient high DTG concentrations initially after dosing. Dosage form adjustments for systemic administration within the population model provided human clearance (CL) estimates. These were similar to what is determined by allometric scaling. This was developed from CL values in mice and rats taken from the present study. Figure 5 show allometric relationships for DTG CL from the US FDA Summary Basis of Approval and those determined in this study. The overall release of DTG following NM2DTG administration in non-human primates is shown as percentage remaining to be absorbed plots and suggested that after an initial equilibrium, the release or appearance of DTG showed first-order (approx. release $t_{1/2} \sim 100$ days) following NM2DTG injection (**figure 5 response**). Also shown is the DTG release profile from NM2DTG in the rats and was similar to rates noted in the non-human primates. Human DTG PK parameters of CL and volume of distribution (V) with DTG release rates, both including inter-subject variability, were used to simulate DTG plasma concentrations (median, 5 and 95 percentiles). Simulations were based on a 70 kg 40-year old subject for 500 simulated subjects with 170 times points generated per year. Simulations following a single 45 mg/kg NM2DTG dose is presented in **figure 5 response**. All together this model predicts that a 50 mg/kg DTG-eq maintenance dose can readily be administered beyond 6 months (**figure 6 response**). This would provide protective plasma levels of DTG 4XPA-IC₉₀ in human subjects at the 95 percentile throughout a 6 month dosing interval. Such a 50 mg/kg DTG-eq dose would translate in a 60 kg human to a ~5 g dose which could be given as two 2.5 g/5 mL injections. These dosing recommendations are expected to be higher since they consider inter-subject variability in PK and absorption. This is given in context of having the 95 percentile of subjects exceed 256 ng/mL DTG plasma concentrations.

Figure 5 response. These graphs show the allometric clearance relationship; First-order release of DTG from NM2DTG in NHPs and rats; predicted median DTG plasma concentrations and the 5 and 95 percentile intervals for simulations made at 170 times over 364 days for 500 simulated human subjects of weight 70 kg and age 40 year and a dose of 45 mg/kg (3150 mg).

Figure 6 response. This graphs show the predicted median (-) DTG plasma concentrations and the 5 (-) and 95 (-) percentile intervals for simulations made at 170 times over 364 days for 500 simulated human subjects of weight 70 kg and age 40 year: 45 mg/kg NM2DTG and a maintenance dose of 50 mg/kg NM2DTG delivery for 6 months

We provide a more simplified allometric scaling model based on our preclinical data which further supports the feasibility of ultra-long-acting dosing intervals in humans. Here, PK parameters for apparent DTG exposure were scaled from observed parameters in our preclinical studies, as well as published data for parent DTG from the published US FDA Summary Basis of Approval, to extrapolate human clinical dose required to achieve ultra-long-acting dosing intervals. Namely, the release of DTG from NM2DTG was described as a first-order-process based on our observed preclinical data (figure 5 response), PK parameters such as clearance (CL) and volume of distribution (V) for NM2DTG were allometrically scaled from our observed data and published data for parent DTG, and published human population PK models of DTG (Br J Clin Pharmacol. 2015 Sep;80(3):502-14). This was used to provide 5% and 95% bounds for simulated population PK. All together the model predicts that a 45mg/kg DTG-eq dose would readily provide protective plasma levels of DTG 4x PA-IC₉₀ throughout 6-month dosing. Such a 45mg/kg DTG-eq dose would translate in a 60 kg human to a ~4g dose which could be given as two 2g 5mL injections. We note that each modeling method offers a different approach with independent predictive analyses. Amongst the three, the MBPK model accurately predict our NHP PK data sets and accounts for the multi-phasic time course of the extended-release prodrug nanocrystals. We note that the human population-based models do not consider initial early DTG concentrations. In addition, this approach would be expected to suggest higher doses since dose selection is based on 95% of subjects having DTG plasma concentrations greater than 256 ng/ml. We also note that simulations of animal data and subsequent BSA scaling factors may be limited in their precision, as the simulations are based on limited datasets in a small number of animals, and we use a single scaling factor rather than scaling individual parameters. Nonetheless, each of these methods on their own support the feasibility of ultra-long-acting dosing intervals for NM2DTG, and all together provide even stronger evidence in support of NM2DTG as an ultra-long-acting product. Given the

limitations discussed, we chose to share the data sets with you and the reviewers at this time rather than place them into the revised text. Within the next year our extended PK analyses that include dose escalation studies, required for a filed IND, will provide the simulation precisions for a complete publication around human dose projection. **Figures 4 and 7** illustrate long-lasting drug depots at the muscle injection site and sustained drug levels in lymphoid tissues. We also note that at the site of viral replication (lymphoid tissues) and site of injection (muscle) drug levels greater than 30x the 4x PA-IC₉₀ at our final measurement time point. Hence, although our NHP studies did not demonstrate 4x PA-IC₉₀ in plasma at the final time point, significant drug levels remain at the relevant tissue site. Furthermore, we note that significant differences have been noted in exposure between NHP and human for integrase inhibitors of similar structure, which may be due to differences in glucuronidation across species. The microsomal enzyme UDP-glucuronosyltransferase can affect blood drug concentrations in pre-clinical studies (Drug Metab Pharmacokinet. 2018 Feb; 33(1): 9–16). As such, non-human-primates can significantly underestimate human exposures for integrase inhibitors such where a 50 mg/kg NHP dose produced exposures similar to 7.5 mg/kg in humans (Sci Transl Med. 2015 Jan 14;7(270):270ra4. doi: 10.1126/scitranslmed.3010298).

Point 2. *“Will the increased dose of the nanoformulation in this technology result in the increase of drug in plasma or will it extend the delivery time?”*

Response: In a replicate report we have shown that the increased dose of nanoformulation in this technology has been shown to increase drug in plasma. These studies were performed in a contract research laboratory blinded to the sample drug designations (Nat Commun. 2021 Jun 8;12(1):3453. doi: 10.1038/s41467-021-23668-x.). The higher dose maintained higher plasma drug levels throughout the length of the study and increased the amount of time drug levels remained above 4x PA-IC₉₀.

Point 3. *“PK studies are using only one sex (male mice and rats, female NHP). Is there any reason for this? Please, justify in the method section.”*

Response: We have now updated the rationale for the sex choices in the methods section. The sex choices were based on the notion that female hormones may cloud the PK data sets and that estrous cycles might cause data variability. However, in the NHPs we wish to confirm these rodent data sets as both males and females will be tested. We plan to investigate NM2DTG PK across species and sexes in future studies. This will be included in our IND-enabling study package to support first-in-human clinical trials.

Point 4. *“The prodrug presence at the injection site allows a convenient drug removal if and when warranted for secondary adverse reactions” is not supported with any data and should be removed. Intramuscular depots of nano formulations are very difficult to remove without muscular damage. In addition, this statement contradicts several results of the study: Figure 2B shows long retention in macrophages (30 days) that would not be affected by the removal of the depot from injection site. Figure 6 d, e, f shows nanocrystals in endosomes of multiple organs, not at the injection site.”*

Response: We have deleted the sentence that refers to injection site removal. Nonetheless, we added **figure 7** which shows DTG prodrug nanocrystal presence at the injection site in monkeys. On balance while it is now supported with the data for long retention in macrophages and now demonstrated at the site of injection, a burden remains on how this may be removed. While levels seen at the injection site do provide a stable depot of the prodrugs and that drug particles are maintained in macrophages safely for a year or longer, removal will require further study. We do affirm that the formulations show no recognized untoward injection site reactions, and these were evaluated throughout the study investigation. No erythema, swelling, and histological reactions were recorded after tissue examination by a blinded pathological assessment made by an experienced pathologist. This includes examination of relevant lymphoid tissues. As the nanoformulations may be difficult to remove without muscular damage the sentence in question was deleted.

Point 4. *“Fig 2 has 1 μm and 10 μm instead 1 μM and 10 μM ”*

Response: Corrected.

Point 5. *“In Figure 2 D-O, please identify nanocrystals and endosomes for clarity as in Supp. Figure 12.”*

Response: Done. Arrows now show the identified nanocrystals.

Reviewer 2

Overall. *“Deodhar et al. created and screened a library of monomeric and dimeric dolutegravir (DTG) prodrug nanoformulations. They identified a prodrug nanocrystal formulation (NM2DTG) that exhibited a far slower decay rate than DTG and maintained levels above 1x PA-IC₉₀ for up to a year in mice and rats. The authors show that the muscle injection site and lymphoid tissues were depots of prodrug hydrolysis and characterize various factors that affect decay rate. Overall, the data on NM2DTG look very promising and support its ultralong acting profile. The authors indicate PrEP as a main use for NM2DTG as it provides a much longer dosing period than the recently approved bimonthly long acting cabotegravir (CAB LA). This work is similar to a recent report by the same group on a CAB prodrug nanocrystal formulation.”*

Response: Agree and thank you very much.

Point 1. *“As the authors acknowledge, the CAB LA trial results demonstrate that plasma concentrations equivalent to 4x PA-IC₉₀ are a target benchmark for PrEP protection. Given the high similarities between CAB and DTG, the authors may want to explicitly indicate this benchmark as a target profile instead of “exceeding the 1x PA-IC₉₀ for one year” stated in the Introduction. Certainly, the authors can adjust this benchmark if they generate data from challenge studies in macaques supporting a lower threshold for complete protection. However, in the absence of such data the 4x PA-IC₉₀ in macaques should remain the working target level. Because the macaque model has now been found to be predictive of clinical efficacy, data from macaques receive heightened attention.”*

Response: Understood and agree. Please see response to the editors (overall).

Point 2. *“While the authors are to be complimented for the comprehensive characterization of NM2DTG, the PK data in macaques dosed with 45 mg DTG-eq./kg showing plasma concentrations falling rapidly under 4x PA-IC₉₀ after the initial and the booster dose on day 217 are a bit underwhelming. The authors do not show any data from dose escalation studies in macaques that demonstrate maintenance of DTG above 4x PA-IC₉₀ for an extended period of time that approaches a year and would translate to a year in humans. The authors singled out rat data for dose extrapolation to humans which may be less convincing than macaque data.”*

Response: Agree in principle. As noted in response above to the editors and other reviewer we now share additional PK modeling to support the feasibility of 4x PA-IC₉₀ in humans for extended dosing intervals, and further demonstrate significant tissue (lymphoid and at the muscle injection site) drug levels well above the protective levels and depot formation. While we acknowledge that the drug levels in plasma are low in NHPs at extended time points, this is not true at the regional lymphoid tissues which are the sites of viral growth, and furthermore we note that NHP observations may significantly underestimate human exposures as recorded for CAB (Spreen et al. 2014 Sci. Trans Med.). The current approach is a paradigm shift certainly in treatment and must be affirmed in patient phase I investigations.

Point 3. *“Other comments:*

- Please add needle size used for the IM injection, and comment on viscosity of the formulation
- Line 262: clarify if the IM injection was done in the same or opposite quadricep

- Line 396: Clarify the H&E was done in rats
- Figure 3C is missing the dotted line for 4x PA-IC₉₀
- Line 441: are drug levels in all lymph nodes similar (mesenteric vs axillary or inguinal? Do the nodes closer to the injection site have higher drug exposures?)
- Line 450: describe what modest metabolic differences were observed.”

Response: Thank you and for each of these points the methods and results were corrected to provide these important and necessary details in experimental design and analysis. To your point about Line 441; with the rats, the lymph nodes were pooled and processed for drug levels together due to sample size limitations. This makes it unknown the contribution of each individual lymph node. However, with newly added tissue drug levels from the monkeys, individual lymph node samples were run and a clear association between distance from the site of injection and drug levels (especially prodrug levels) is seen.

Reviewer 3

Overall. *“The study in the manuscript, “Pathways for Dolutegravir Transformation from a Daily Oral to a Once-a-Year Parenteral Medicine” is among the most important finding, to date, that describes a pathway to effectively treat HIV. The information presented here is built upon the notion that although ART drugs are available for all the HIV-infected individuals, the treatment is not very effective for a variety of reasons. For examples, lack of medication adherence, toxicity, drug interactions, social stigma, prevalence of drugs of abuse, and behavior. However, if the burdens of the ART drug regimen are reduced from daily or weekly to monthly or even yearly, then the treatment could be highly effective in HIV populations. Over the years, the scientific community has been working on finding a long-acting ART drug for HIV treatment. In this respect, the finding here strongly provides the evidence that dolutegravir, an integrase inhibitor, can be used as intramuscular depot that ultra-slowly releases the drugs up to a year. In this context, the study is highly significance in the HIV field. The work is also original and results clearly support the conclusions. This will significantly advance the field. The methodologies in this manuscript is also robust, because the study not only synthesized nano-prodrugs and extensively characterized them, but also used in-vitro primary macrophages and three appropriate animal models, mice, rat, and rhesus to characterize the drug formulations. They are also described well. Although the number of biological replicates came from fewer cells or animals (mostly 3-4), the data from the final formulation NM2DTG, compared to others, are so significant and noticeable that, additional biological replicates would not offer any additional power or significance. The manuscript is well-written in a lay-men language and builds an impressive story of its future application upon clinical studies in humans. The overall data, both in manuscript main figures and supplemental figures are all clean, well-presented, and comprehensive. The statistical analyses are also appropriate.”*

Response: Agreed and thank you so very much.

Point 1. *“There is no explanation, if relevant, why rats were male and rhesus were female. Both male and female are important to be considered in the study, unless its justified. It’s understood that with 3-4 animals, it’s hard to choose equally both the sexes.”*

Response: Understood and corrected with an explanation provided in the methods section.

Point 2. *“In results section, fig 1c does not have PK profile, however, its mentioned in the text, “drugs affect both the PK and aqueous solubility.”*

Response: Thank you and the text was corrected.

Point 3. *“In results section, the description of fig 3c comes after fig 3d-g. It’s important to describe the figure in the text in the same order that they are presented in the figure.”*

Response: We made the needed revisions so that the text is now aligned with the figure.

Point 4. *“Figure 6 is an important summary figure. However, its presentation appears to be missing in the discussion section.”*

Response: Figure 6 (now Fig. 8) provides a necessary overview for the novel approach offered in this report. We have expanded the discussion section, so its presentation is highlighted.

We affirm that the data sets are of immediate impact as a safe, year-long agent that could have a substantive effect on preventing HIV transmission, and hence on the pandemic. The interest in year-long acting antiretrovirals is undeniable from both prevention and eradication paradigms. We wish to thank you and the reviewers for your thoughtful comments. Please do not hesitate to call on me once again if any further needs are required.

With best regards,

Howard E. Gendelman, M.D.

Margaret R. Larson Professor of Internal Medicine and Infectious Diseases

Professor and Chairman, Department of Pharmacology and Experimental Neuroscience

REVIEWERS' COMMENTS

Reviewer #1 (Remarks to the Author):

The authors have satisfactorily addressed my comments and concerns.

Reviewer #2 (Remarks to the Author):

The authors addressed adequately my comments. I recommend acceptance.

May 12, 2022

Re: NCOMMS-22-01074 "Transformation of Dolutegravir to an Ultra-Long-Acting Parenteral Prodrug Formulation"

Thank you for the thorough and reasoned review of our manuscript. We are pleased to respond to each of the queries raised point-by-point.

Reviewer 1

The authors have satisfactorily addressed my comments and concerns.

Response: Thank you so very much

Reviewer 2

The authors addressed adequately my comments. I recommend acceptance.

Response: Agreed and thank you so very much.

Reviewer 3

N/A

We affirm that the data sets are of immediate impact as a safe, year-long agent that could have a substantive effect on preventing HIV transmission, and hence on the pandemic. The interest in year-long acting antiretrovirals is undeniable from both prevention and eradication paradigms. We wish to thank the reviewers for their time and thoughtful comments.

With best regards,

Howard E. Gendelman, M.D.

Margaret R. Larson Professor of Internal Medicine and Infectious Diseases
Professor and Chairman, Department of Pharmacology and Experimental Neuroscience